# Characterization of an alternative BAK-binding site for BH3 peptides

Kaiqin Ye[1,2], Wei X. Meng[3,4], Hongbin Sun [5], Bo Wu[6], Meng Chen[1,2], Yuan-Ping Pang [4], Jia Gao[1,2], Hongzhi Wang[1,2], Junfeng Wang [6], Scott H. Kaufmann [3,4✉] & Haiming Dai [1,2✉]

Many cellular stresses are transduced into apoptotic signals through modification or up-regulation of the BH3-only subfamily of BCL2 proteins. Through direct or indirect mechanisms, these proteins activate BAK and BAX to permeabilize the mitochondrial outer membrane. While the BH3-only proteins BIM, PUMA, and tBID have been confirmed to directly activate BAK through its canonical BH3 binding groove, whether the BH3-only proteins BMF, HRK or BIK can directly activate BAK is less clear. Here we show that BMF and HRK bind and directly activate BAK. Through NMR studies, site-directed mutagenesis, and advanced molecular dynamics simulations, we also find that BAK activation by BMF and possibly HRK involves a previously unrecognized binding groove formed by BAK α4, α6, and α7 helices. Alterations in this groove decrease the ability of BMF and HRK to bind BAK, permeabilize membranes and induce apoptosis, suggesting a potential role for this BH3-binding site in BAK activation.

[1] Anhui Province Key Laboratory of Medical Physics and Technology, Center of Medical Physics and Technology, Hefei Institutes of Physical Science, Chinese Academy of Sciences, Hefei 230031, China. [2] Hefei Cancer Hospital, Chinese Academy of Sciences, Hefei 230031, China. [3] Division of Oncology Research, Mayo Clinic, Rochester, MN, USA. [4] Department of Molecular Pharmacology and Experimental Therapeutics, Mayo Clinic, Rochester, MN 55905, USA. [5] School of Food and Biological Engineering, Zhenzhou University of Light Industry, Zhenzhou 450002, China. [6] CAS Key Laboratory of High Magnetic Field and Ion Beam Physical Biology, High Magnetic Field Laboratory, HFIPS, Chinese Academy of Sciences, Hefei 230031, China. ✉email: Kaufmann.Scott@mayo.edu; daih@cmpt.ac.cn

BCL2 family proteins regulate the mitochondrial apoptosis pathway by controlling mitochondrial outer membrane (MOM) permeabilization[1–3]. The BCL2 protein family is divided into three subfamilies: antiapoptotic BCL2 family proteins, including BCL2, BCLx$_L$, and MCL1; proapoptotic multidomain BCL2 family proteins, including BAK, BAX, and BOK, which can directly permeabilize the MOM after activation; and proapoptotic BH3-only proteins, including BIM, PUMA, BID, BAD, NOXA, BMF, HRK, and BIK, which share homology with other BCL2 family members only in a 15- to 25-residue BH3 domain.

BH3-only proteins sense various cellular stress signals and transform them into death signals[4,5]. For example, endoplasmic reticulum (ER) stress causes BIM upregulation and subsequent cell death[6]; BMF transforms signals arising during cellular detachment or actin depolymerization to apoptosis[7]; and HRK converts growth factor withdrawal signals to cell death in neurons[8]. Another BH3-only protein, BIK, also transduces signals initiated by various anticancer agents, including DNA cross-linking agents[9], lenalidomide[10], and anti-BCR (B-cell receptor) therapy[11], into cell death signals. Accordingly, elucidating the action of these BH3-only proteins is important for understanding how various treatments induce cell death.

BH3-only proteins can induce mitochondrial apoptosis in two different ways: (1) By displacing activated BAK/BAX from anti-apoptotic BCL2 family members[12,13] and (2) by directly interacting with and activating BAK/BAX[14–16]. Depending on their ability to directly activate BAK/BAX or not, BH3-only proteins are divided into direct activators, which can directly bind BAK and/or BAX to induce their conformational change, oligomerization, and activation[15,17,18], and sensitizers[15,19], which inhibit antiapoptotic BCL2 family proteins to release the direct activators or activated BAK/BAX[20,21].

Experiments examining which BH3-only proteins are direct activators and which are sensitizers have yielded divergent conclusions. BIM and tBID were originally reported to be the only activators[15,19]. PUMA was subsequently suggested to be a direct activator[22–24], albeit with lower potency than BIM[25]. Whether BIK, BMF and HRK are direct activators is less clear. The original experiments identified these three BH3-only proteins as sensitizers[15,19], but later studies suggest these proteins can also activate BAK or BAX, again with lower potency[26]. Moreover, a recent study using chimeras in which the BID BH3 domain is replaced by BH3 peptides from other BH3-only proteins also indicates that BMF, HRK and BIK can activate BAK/BAX[27], but this conclusion disagrees with results in a yeast model system that questions whether BIK and BMF act as direct activators[28].

A number of studies have found that BIM, PUMA, and tBID interact with the canonical BH3-binding groove formed by BAK α3, α4, and α5[17,25,29,30]. However, the interaction surfaces of BMF and HRK on BAK have not been reported. In this study, we evaluate the abilities of BMF, HRK, and BIK to directly interact with BAK and induce BAK-mediated membrane permeabilization. Moreover, using nuclear magnetic resonance (NMR) spectroscopy, molecular dynamics (MD) simulations and site-directed mutagenesis, we map the sites where BMF and HRK interact with BAK. Interestingly, these studies suggest that activation of BAK by BMF and possibly HRK involves binding to a previously unrecognized heterodimerization groove on BAK.

## Results

**Direct binding of BMF and HRK BH3 peptides to BAK**. We have previously shown BIM, NOXA, tBID, and PUMA can directly bind and activate BAK[17,25]. Because all known direct activator BH3-only proteins interact with BAK through their BH3

domains[15,30], we first asked whether synthetic BMF, HRK, and BIK BH3 peptides (Fig. 1a) can interact with BAKΔTM. BIM and PUMA BH3 peptides served as positive controls, whereas BAD BH3 peptide (the prototypic sensitizer) served as a negative control[25].

Surface plasmon resonance (SPR) indicated that the BMF BH3 domain binds to BAKΔTM with a dissociation constant ($K_D$) of <1.0 μM (Figs. 1b, f and Supplementary Fig. 1a), which is smaller than the $K_D$ of BIM BH3 with BAKΔTM (Figs. 1d, f and Supplementary 1c). The HRK BH3 peptide binds to BAKΔTM with $K_D$ in the low micromolar range (Figs. 1c, f and Supplementary Fig. 1b), which is similar to the affinity of BIM and PUMA BH3 peptides for BAKΔTM (Figs. 1d, f and Supplementary Fig. 1c, d)[26,29–31]. In contrast, BAD BH3 domain showed a much lower affinity (Fig. 1f). Direct comparison of the SPR curves (Fig. 1g) showed that BMF BH3 exhibited the fastest association with BAKΔTM and slowest dissociation, resulting in the smallest $K_D$.

Surprisingly, SPR did not detect interactions between the BIK BH3 peptide and BAKΔTM (Fig. 1e–f). Circular dichroism spectroscopy revealed similar α-helicity of the BH3 peptides (Supplementary Fig. 2a, b), ruling out the possibility that the differences in $K_D$s reflected differences in ability of these peptides to form secondary structures.

**BMF and HRK BH3 domains induce BAK activation in vitro**. To determine whether BMF and HRK, like BIM, tBID, PUMA, and NOXA[17] can induce BAK activation, we initially examined BAK-mediated liposome permeabilization. As shown in Figs. 2a, b, addition of the BMF or HRK BH3 increased BAK-dependent release of liposome contents from 28% (BAK alone) to 46% (BMF, $p < 0.01$) and 48% (HRK, $p < 0.001$, two-tailed paired $t$ test), respectively. These results are slightly lower than the BIM BH3 (67% release) but similar to BID BH3 (50%). In contrast, BIK BH3 and the negative control BAD BH3[17,25–27,32] did not appreciably increase BAK-mediated liposome release.

To further evaluate the ability of these peptides to activate BAK, we studied cytochrome c release using mitochondria from $Bax^{-/-}$ mouse embryonic fibroblasts (MEFs) (Supplementary Fig. 3a). Treatment with the BCL2/BCLx$_L$ inhibitor navitoclax, the MCL1 inhibitor S63845, or combination of the two failed to induce cytochrome c release, suggesting that Bak is not autoactivated[21] in these mitochondria (Figs. 2c, d and Supplementary Fig. 3b). Consistent with these results, BH3 peptide from the sensitizer protein BAD also did not induce cytochrome c release even at 10 μM (Fig. 2e). In contrast, BIM BH3 peptide, which reportedly has the strongest ability to activate Bak[27], induced near maximal cytochrome c release at 0.2 μM; and BID BH3 peptide-induced cytochrome c release at 2 μM (Fig. 2e). BMF BH3 peptide-induced cytochrome c release at concentrations similar to the BID BH3, whereas the HRK and BIK peptides only induced cytochrome c release at 10 μM, the highest concentration tested.

Because cytochrome c release from $Bax^{-/-}$ mitochondria did not exactly match the liposome permeabilization results shown in Figs. 2a, b, we hypothesized that BH3 peptide-induced cytochrome c release from mitochondria of $Bax^{-/-}$ MEFs might reflect a combination of direct Bak activation and antiapoptotic paralog neutralization. To test this idea, S63845 or navitoclax was applied to the mitochondria along with BIK BH3 or HRK BH3. In the presence of S63845, HRK BH3-induced cytochrome c release at a concentration 10 times lower than BIK BH3 (Fig. 2f), in agreement with the binding and the liposome permeabilization results. In contrast, navitoclax did not enhance either BIK or HRK BH3-induced cytochrome c release (Supplementary Fig. 3c), suggesting that Bak was predominantly bound to Mcl1 after activation.

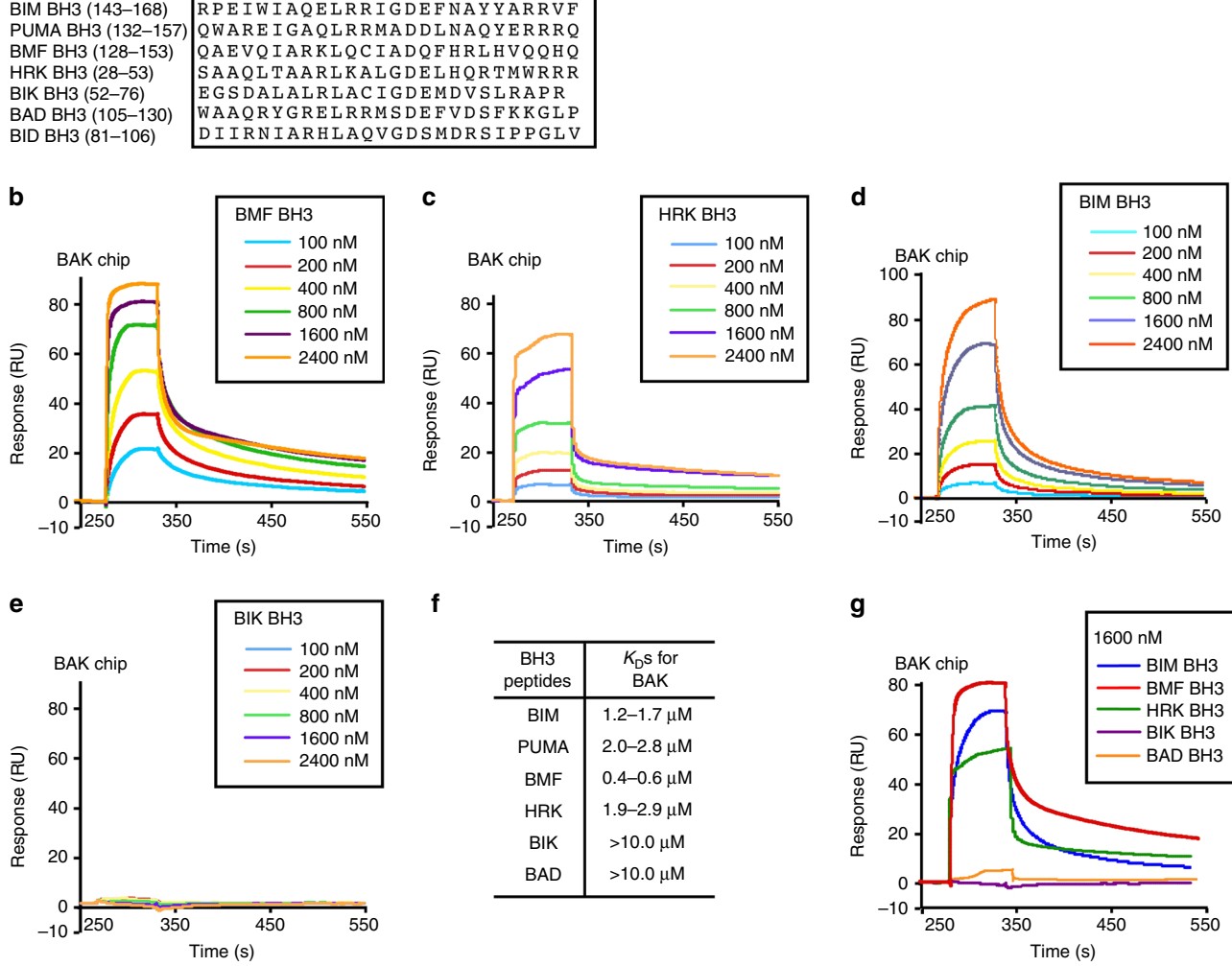

**Fig. 1 Direct interaction of BMF and HRK BH3 peptides with BAKΔTM. a** Synthetic BH3 peptides used in this study. **b–e** SPR (relative units (RU)) observed when immobilized BAKΔTM was exposed to a series of concentrations of BMF BH3 **b**, HRK BH3 **c**, BIM BH3 **d**, or BIK BH3 **e**. **f** Approximate dissociation constant ranges of BAKΔTM interacting with the indicated BH3 peptides estimated from three independent SPR assays using steady-state analysis (Supplementary Fig. 1). Because artefacts were observed when high concentration of certain peptides were applied, the determination of BAK:BH3 peptide affinities did not meet the criterion required for steady-state analysis that top concentrations should be at least threefold above the reported $K_D$. **g** Direct comparison of BAKΔTM interacting with the indicated BH3 peptides at 1600 nM. $K_D$ dissociation constant. Source data are provided as a source data file.

We also examined cytochrome c release by incubating mitochondria from $Bak^{-/-}Bax^{-/-}$ MEFs with BH3 peptides plus purified human BAKΔTM. Under these conditions, where more recombinant BAK was present, BIM BH3 also induced the most cytochrome c release, BMF- and HRK-induced release of an intermediate amount, and BIK BH3 barely induced cytochrome c release (Fig. 2g). Collectively, these results indicate that under cell-free conditions BIM BH3 peptide is the strongest BAK activator tested, BMF BH3 and HRK BH3 are weaker, and BIK BH3 is barely able to activate BAK.

**BMF and HRK induce direct BAK activation in cells.** To assess Bak activation by full-length BMF and HRK proteins, we measured induction of apoptosis by BMF and HRK. In initial experiments, wild type (WT) MEFs were transfected with EGFP-tagged BH3-only proteins (Supplementary Fig. 4a), incubated for 24 h, and stained with Annexin V to assess phosphatidylserine externalization in EGFP+ (successfully transfected) cells. Typically, >60% cells were EGFP+ 24 h after transfection. In the WT

MEFs, EGFP-tagged BIM, PUMA, BMF, and HRK all induced apoptosis in >75% of EGFP+ cells, suggesting potent proapoptotic action of these BH3-only proteins in situ. In contrast, BAD, which is not an activator of either BAK or BAX, only induced apoptosis in ~25% of WT MEFs. Because of poor BIK expression in the MEFs (Supplementary Fig. 4a), results with this protein were inconclusive. Upon transfection into $Bak^{-/-}Bax^{-/-}$ MEFs, none of these proteins induced apoptosis (Supplementary Fig. 4b), confirming that cell death induced by these BH3-only proteins depends on Bak or Bax.

Using MEFs with only Bax ($Bak^{-/-}$ MEFs) or only Bak ($Bax^{-/-}$ MEFs), no obvious preference for Bak or Bax was found after transfection with BMF, HRK, BIM, PUMA, or tBID (Supplementary Fig. 4c, d). NOXA, however, induced more apoptosis in $Bax^{-/-}$ than $Bak^{-/-}$ MEFs, suggesting differential ability of NOXA to activate Bak (Supplementary Fig. 4c, d).

In Jurkat cells, which express ~8 times more BAK than BAX[33], BMF and HRK also induced apoptosis in 83% and 89% of successfully transfected cells, comparable to the direct BAK activator BIM and different from the BAK sensitizer BAD

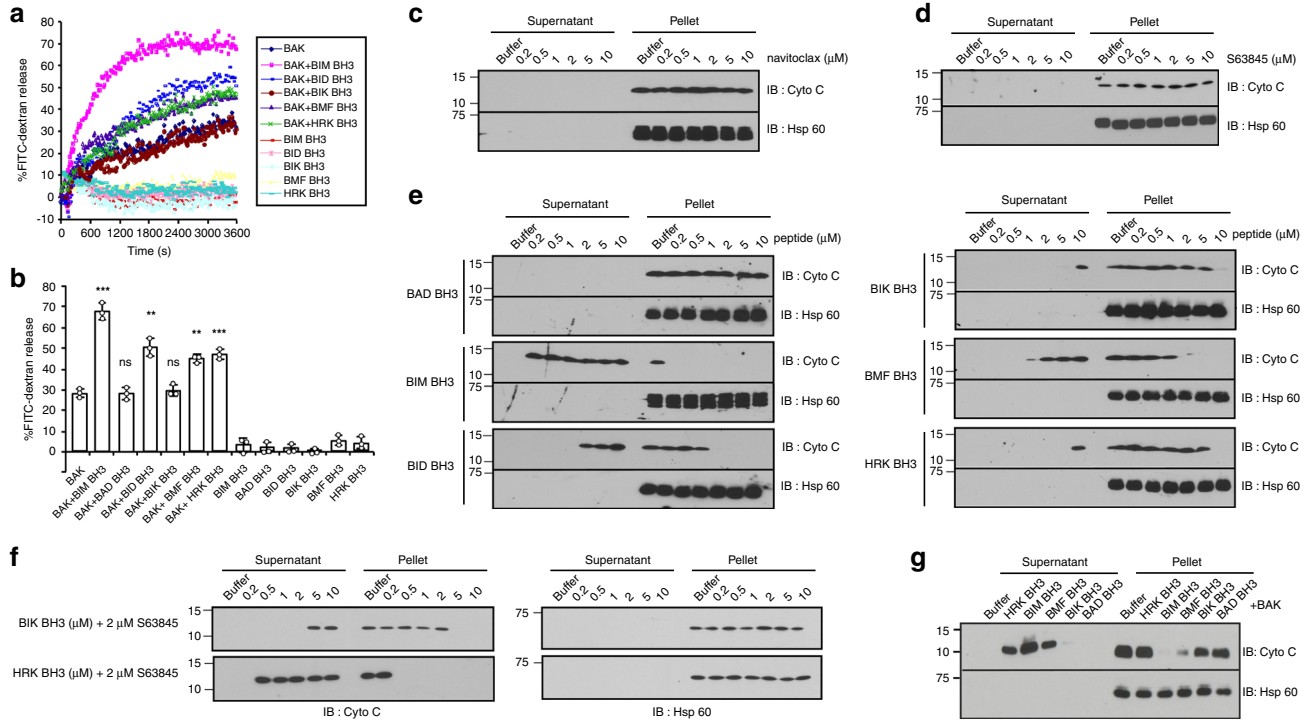

**Fig. 2 BMF and HRK BH3 peptides directly activate BAK. a, b** Liposome permeabilization assay performed in the presence of 50 nM BAK and/or 50 nM of the indicated BH3 peptides. A representative experiment **a** and summary of the percentage of FITC-dextran release **b** are shown. Error bars: mean ± S.D. of three independent experiments. ***$p < 0.001$; **$p < 0.01$; ns, not significant; two-tailed paired $t$ test, ($p = 0.0008$, 0.82, 0.002, 0.39, 0.001, and 0.0004, for BAK + BH3 peptides of BIM, BAD, BID, BIK, BMF, or HRK BH3 peptide vs. BAK alone, respectively). **c–f** After mitochondria from $Bax^{-/-}$ MEFs were incubated for 90 min at 25 °C with the indicated concentrations of navitoclax **c**, S63845 **d**, or multiple BH3 peptides **e**, or BIK or HRK BH3 peptide plus S63845 **f**, supernatants and pellets were subjected to SDS-PAGE and immunoblotting. **g** After mitochondria from $Bax^{-/-}Bak^{-/-}$ MEFs were incubated for 90 min at 25 °C with purified BAKΔTM together with the indicated BH3 peptides, supernatants and pellets were subjected to SDS-PAGE and immunoblotting. Numbers on the left side of **c–g** and western blots in other figures indicate migration of the molecular markers. Cyto c, cytochrome c. Source data are provided as a source data file.

(Fig. 3a, b). Moreover, EGFP-BIK also induced apoptosis in 85% of EGFP⁺ cells, perhaps reflecting the fact that BAK is autoactivated in Jurkat cells[21] and can be displaced from BCLxₗ and MCL1 by BIK. This apoptosis was accompanied by BAK activation, as detected by an antibody that recognizes an active BAK conformation (Fig. 3c)[17,25] and abolished by *BAK* knockout (Fig. 3d), consistent with the important role of BAK in apoptosis induction in these cells.

To further explore BMF- and HRK-induced BAK activation without the confounding effects of known endogenous BH3 activators and BAK autoactivation, we generated *BIM⁻/⁻ PUMA⁻/⁻BID⁻/⁻NOXA⁻/⁻BAK⁻/⁻BAX⁻/⁻* Jurkat cells using CRISPR-Cas9 technology (Fig. 3e) and stably transfected these HexaKO cells with a plasmid containing BAK behind a doxycycline inducible promoter (Fig. 3f). When BAK was induced by 10 ng/mL doxycycline for 16 h, the BAK concentration was much lower than endogenous BAK levels in WT Jurkat cells (Fig. 3f), and no autoactivated BAK was detectable in BCL2, BCLXₗ, and MCL1 immunoprecipitates (Fig. 3g)[21]. If BMF or HRK were simply inducing apoptosis by displacing activated BAK from antiapoptotic BCL2 family members, there should be no BMF- or HRK-induced apoptosis in this cell line (because there is no pre-activated BAK to displace). However, both BMF and HRK-induced apoptosis in 45–55% of successfully transfected cells in both clones (Fig. 3h, i), suggesting that BMF and HRK can directly activate BAK in this cell line. In contrast, BIK could only induce apoptosis in ~20% of successfully transfected HexaKO cells, again suggesting a much weaker BAK activating ability.

Collectively, results in Fig. 3 indicate that BMF and HRK are sufficient to induce direct BAK activation and BAK-mediated apoptosis in different cell lines.

**A possible second binding site for BH3 peptides on BAK.** Earlier studies identified the canonical BH3-binding groove of BAK as the site for interaction with the BH3 domains of BIM, PUMA, and tBID[17,25,29,30]. To address whether BMF and HRK bind to the same site, an NMR-based chemical shift perturbation assay was used. In brief, ¹⁵N-labeled BAK 15–186 was purified as described in the Methods. After the assignment of the peaks on ¹H, ¹⁵N-HSQC of BAK 15–186 (Supplementary Fig. 5a, b), chemical shift perturbations were measured upon addition of PUMA, BIM, BMF, and HRK BH3 peptides (Fig. 4, Supplementary Fig. 6 and Supplementary Table 1). Because of protein precipitation during incubation with BIM BH3 peptide, we could not obtain high-quality ¹H, ¹⁵N-HSQC perturbation data. Accordingly, PUMA BH3 was used as a positive control. As shown in Fig. 4a and Supplementary Fig. 6b, PUMA BH3 induced two types of chemical shift changes on BAK ¹H, ¹⁵N-HSQC spectrum. One group of residues, including Y89, D90, S91, F93, T95, L97, H99, K113, I114, L118, G126, A130, L132, G133, and F134, exhibited ligand-induced decreases in peak intensity or signal disappearance. Another group of residues, including Q94, Q98, F157, L163, H165, and C166, exhibited focal, dose-responsive chemical shift changes. When mapped onto the structure of BAK (PDB ID: 2IMS[34], Fig. 5a and Supplementary

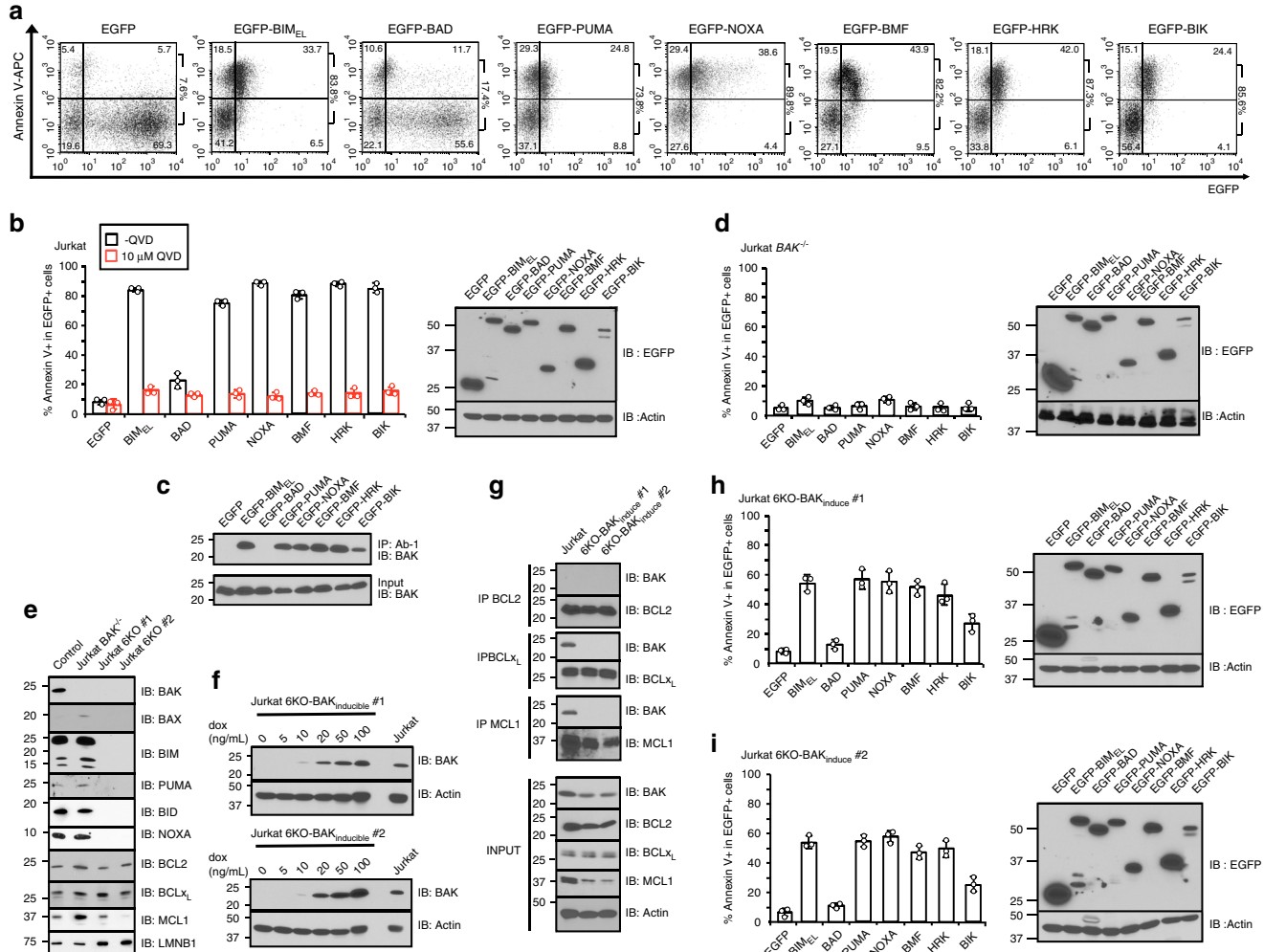

**Fig. 3 BMF and HRK induce BAK/BAX-dependent apoptosis. a**, **b**, **d** After WT **a**, **b** or *BAK*−/− **d** Jurkat cells were transfected with the indicated EGFP-tagged BH3-only proteins in the absence or presence of Q-VD-OPh for 24 h, cells were harvested, stained with APC-conjugated Annexin V, and subjected to flow cytometry. A representative experiment **a** and summarized results **b**, **d** indicate the percentage of apoptotic cells (Annexin V+) among successfully transfected cells (EGFP+). Numbers at the right of the dot plots in **a**, indicate the percentage EGFP + cells (i.e., successfully transfected) that were Annexin V+, which was calculated as the ratio of events in the upper right to the sum of the lower right and upper right. **c** After WT Jurkat cells were transfected with indicated plasmids in the presence of Q-VD-OPh for 24 h, cell lysates were incubated with BAK Ab-1 antibody. **e** Characterization of *BIM*−/−*PUMA*−/−*BID*−/−*NOXA*−/−*BAK*−/−*BAX*−/− HexaKO Jurkat cells by western blotting. **f** After two clones (#1 and #2) of Jurkat HexaKO (6KO) cells with inducible BAK were treated with the indicated concentrations of doxycycline (dox) for 16 h, whole-cell lysates were probed with BAK antibody. **g–i** After the two clones of Jurkat HexaKO with BAK inducible plasmid were induced with 10 ng ml−1 doxycycline for 16 h, cells were lysed, immunoprecipitated with indicated antibodies and blotted for BAK. Jurkat cells served as a positive control **g**. Meanwhile, both clones **h**, **i** were transfected with EGFP-tagged BH3-only plasmids for 24 h, harvested, and stained with APC-conjugated Annexin V. The percentages EGFP+ cells that were Annexin V+ are indicated. Right panels in **b**, **d**, **h**, **i**, whole-cell lysates subjected to immunoblotting. Error bars: mean ± S.D. of three independent experiments. Source data are provided as a source data file.

Fig. 6b), these residues were mostly located in the canonical BH3-binding groove, in agreement with previous studies[25,29,30]. Interestingly, at the same concentration of BMF or HRK BH3 domain, changes for a group of BAK residues (Y89, D90, S91, F93, T95, and L97 on helix α3 as well as G126, A130, L132, G133, and F134 on helix α5) were absent or much lower than with PUMA. Instead, addition of either the BMF or HRK BH3 peptide-induced prominent chemical shift changes that involved residues M96, Q98, and H99 on BAK helix α3; K113, I114, L118 on α4; F157, F161, M162, L163, H165, and C166 on α6; and A168 on helix α7 (Fig. 4b and Supplementary Fig. 6a, c, d). Most of the residues were situated within an α4/α6/α7 groove on BAK (Fig. 5b, c), raising the possibility that the BH3 domains of BMF and HRK might bind to the α4/α6/α7 groove of BAK, although some of the residues in the canonical groove (Q94, L97, Y108,

L118, G126, L131, and F134) also displayed limited but detectable reduction in peak intensity after addition of BMK or HRK peptides (see Supplementary Fig. 6c–d and Discussion).

To further investigate whether the BMF and HRK peptides preferentially bind the α4/α6/α7 groove or canonical BH3-binding groove, we conducted comprehensive, microsecond, isobaric–isothermal MD simulations of BAK with the BIM, BMF, and HRK BH3 peptides docked at either groove. These simulations used FF12MC, an AMBER protein forcefield with improved effectiveness in simulating localized disorders of folded proteins that has accurately simulated experimental folding times for fast-folding proteins[35,36]. Using the model protein CLN025[37] as a control to validate our conformational stability analysis method, we showed that the most populated conformers at 300 K and 277 K are nearly identical to the experimentally determined

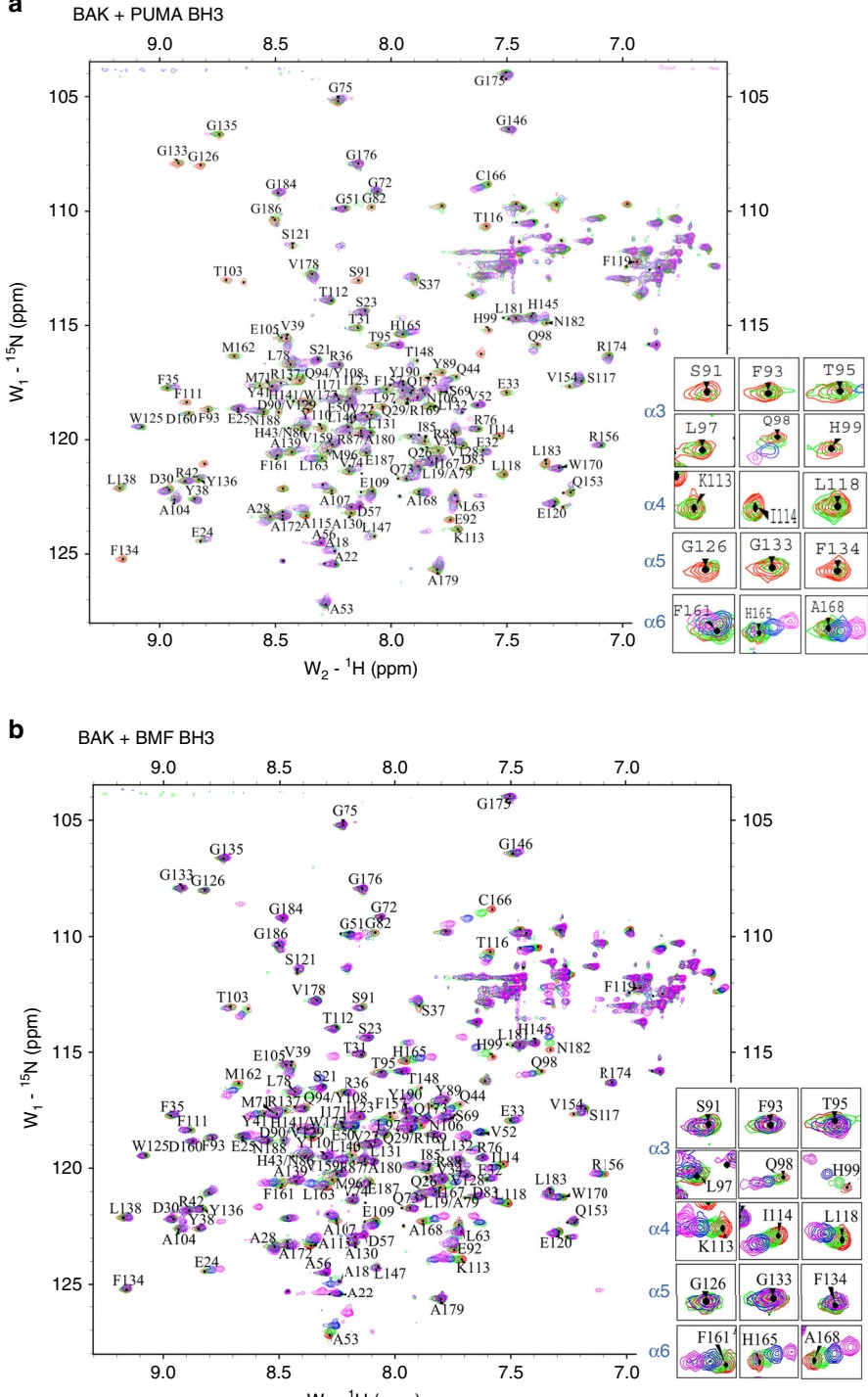

**Fig. 4 1H,15N-HSQCs of BAK 15–186 indicate different residues perturbed by BMF and PUMA BH3. a, b** 1H,15N-HSQC spectra of 0.5 mM 15N-labeled BAK 15–186 perturbed by different concentration ratios of PUMA BH3 peptide **a** or BMF BH3 peptide **b** shown in different colors. Protein:peptide ratios were as follows: Red, no peptides; green, 1:1; blue, 1:2; and purple, 1:4. Peaks on the right indicate the perturbations at selected residues.

CLN025 conformer as indicated by the alpha carbon root mean square deviation of 1.7 Å between the simulated and experimental conformers, and that the CLN025 population increases as the temperature decreases (65% at 300 K and 71% at 277 K as listed in Supplementary Table 2), indicating the correlation of the population with the relative stability of CLN025. Using this method, we obtained the relative populations (Supplementary Table 2) of the most populated BAK complexes with the three BH3 peptides at either groove (Supplementary Fig. 7). These

populations (given in parentheses hereafter) suggest that: (1) BIM BH3 binds with a strong preference to the canonical α3/α4/α5 groove (75%) over the noncanonical α4/α6/α7 groove (35%), in agreement with earlier studies[17,29,30]; (2) BMF BH3 binds both grooves with a possible preference for the noncanonical groove (65% for the canonical and 69% for the noncanonical); and (3) HRK BH3 also binds both grooves (74% for the canonical and 53% for the noncanonical). Despite its higher population in the canonical groove, HRK is thought to also bind the noncanonical

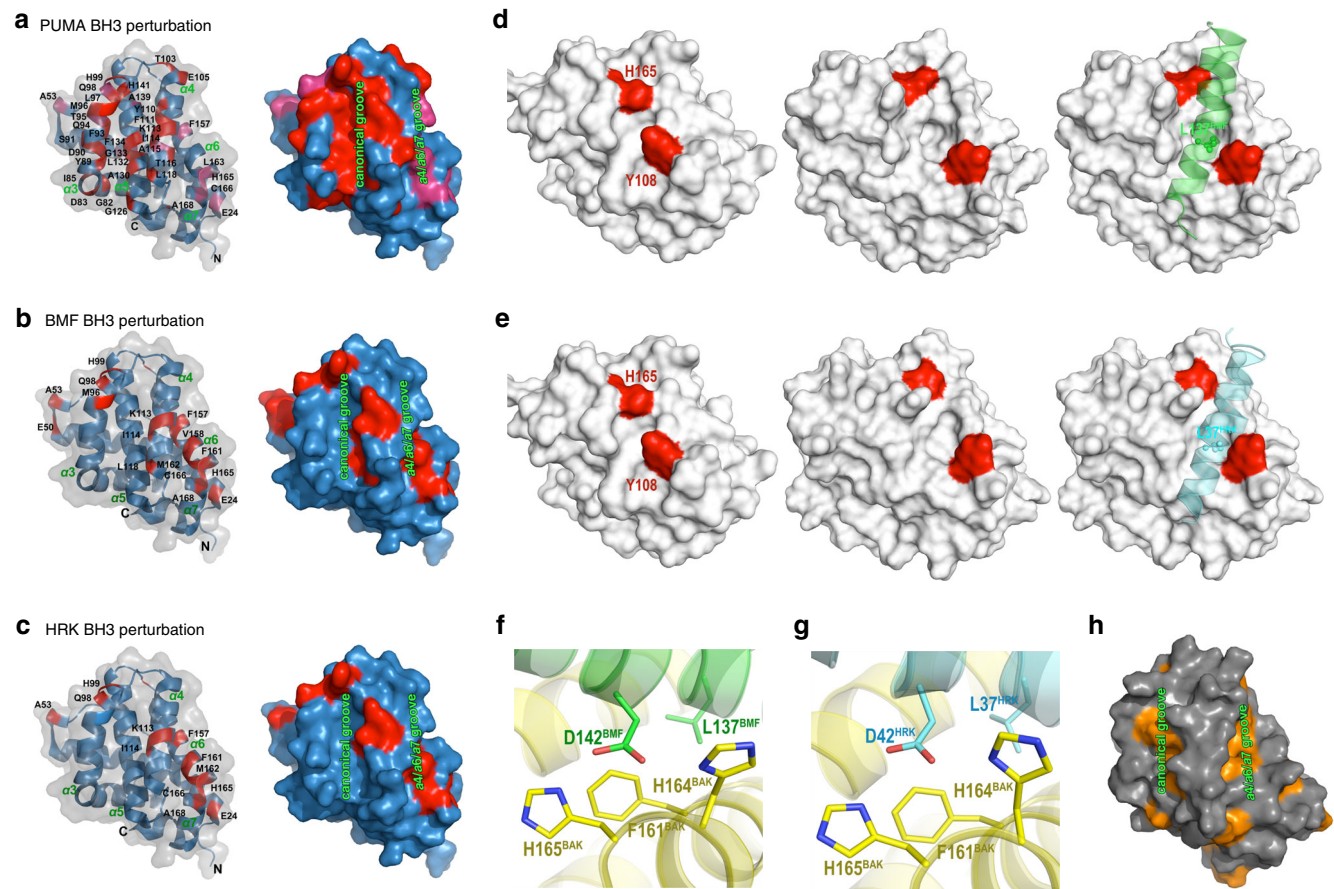

**Fig. 5 BMF and HRK BH3 domains perturb residues that differ from PUMA BH3. a–c** Surface and cartoon mixed model (left) or surface model (right) of BAK (PDB:2IMS) showing residues (indicated in red) that have obvious chemical shift changes or disappearance of signals (peak intensities decreased to the level of noise) on the $^1$H,$^{15}$N-HSQC spectra after the perturbations by PUMA BH3 **a**, BMF BH3 **b**, or HRK BH3 **c**. **d**, **e** Conformational change of Y108 and H165 residues on BAK induced by BMF BH3 **d** and HRK BH3 **e** to expose the α4/α6/α7 groove. Left, before adding peptides; middle, after adding peptides; right, with peptides. **f**, **g** Close-up view of the primary intermolecular interactions of BAK with BMF BH3 **f** and HRK BH3 **g**. **h** Hydrophobic analysis of BAK was generated by Pymol with the overall structure displayed in gray and the hydrophobic surface displayed in orange.

groove based on its high population at the noncanonical groove (53%) relative to BIM at the same site (35%). This involvement of both grooves for BMF and possibly HRK is in agreement with the chemical shift perturbation data (Supplementary Fig. 6c, d).

In addition, the simulations revealed that BAK Y108 and H165 moved out of the α4/α6/α7 groove, exposing a deep hydrophobic cavity to accommodate the conserved L137 of BMF or L37 of HRK, whereas F161 moved up slightly and remained in the groove to (Fig. 5d, e). The simulations also showed that BAK F161 interacted with BMF L137 or HRK L37, whereas BAK H164 and H165 formed salt bridges with the conserved D142 of BMF or D42 of HRK (Fig. 5f, g). A hydrophobic surface analysis indicated that the α4/α6/α7 groove induced by the two peptides was primarily hydrophobic (Fig. 5h). These computational studies are consistent with the NMR results showing prominent chemical shift perturbations along the α4/α6/α7 groove upon binding of BAK to BMF or HRK BH3.

**Effects of BAK and BH3 mutations**. To further assess whether binding of BMF and possibly HRK BH3s involves the α4/α6/α7 groove, SPR was used to assay interactions of BIM, BMF and HRK BH3 peptides with WT or mutant BAKΔTM. Two mutations that physically block the canonical BH3-binding groove (BAK G126S and BAK F93E) diminished binding of BIM BH3 domain (Fig. 6a left and b), in agreement with previous results[17,25,38], but did not appreciably affect binding of BMF and

HRK BH3s (Fig. 6a middle and right, Fig. 6b). Conversely, the helix α6 mutations BAK F161A and H164/H165A diminished binding of the BMF and HRK BH3 domains (Fig. 6c, d) but did not significantly change binding of BIM peptide. These SPR results show that physical blockage of the canonical binding groove affects BIM BH3 binding much more than BMF or HRK binding, consistent with the NMR and MD simulation results suggesting that BMF and HRK can bind at two sites.

We also examined the effect of the BAK F161A mutation using MD simulations. Because α4 separates the α4/α6/α7 and α3/α4/α5 grooves (Fig. 5a–c), the F161A mutation simultaneously contracted the BAK noncanonical groove and expanded the canonical BH3-binding pocket. These changes modestly reduced the population of the BIM BH3 peptide at the canonical groove from 75 to 50% (Supplementary Table 2). However, the populations for BMF binding at the noncanonical and canonical grooves were markedly reduced to 32% and 22%, respectively, in the F161A mutant from 69 and 65% for WT BAK (Supplementary Table 2). Likewise, those of HRK at the noncanonical and canonical grooves of the F161A mutant were also markedly decreased to 22% and 32% from 53 and 74% for WT BAK, respectively (Supplementary Table 2). These simulations suggest that the F161A mutation inhibits binding of BMF and HRK BH3 peptides at both grooves while sparing the binding of BIM at the canonical groove.

To further study these interactions, we also mutated three conserved hydrophobic residues in the BH3 domain of BMF or

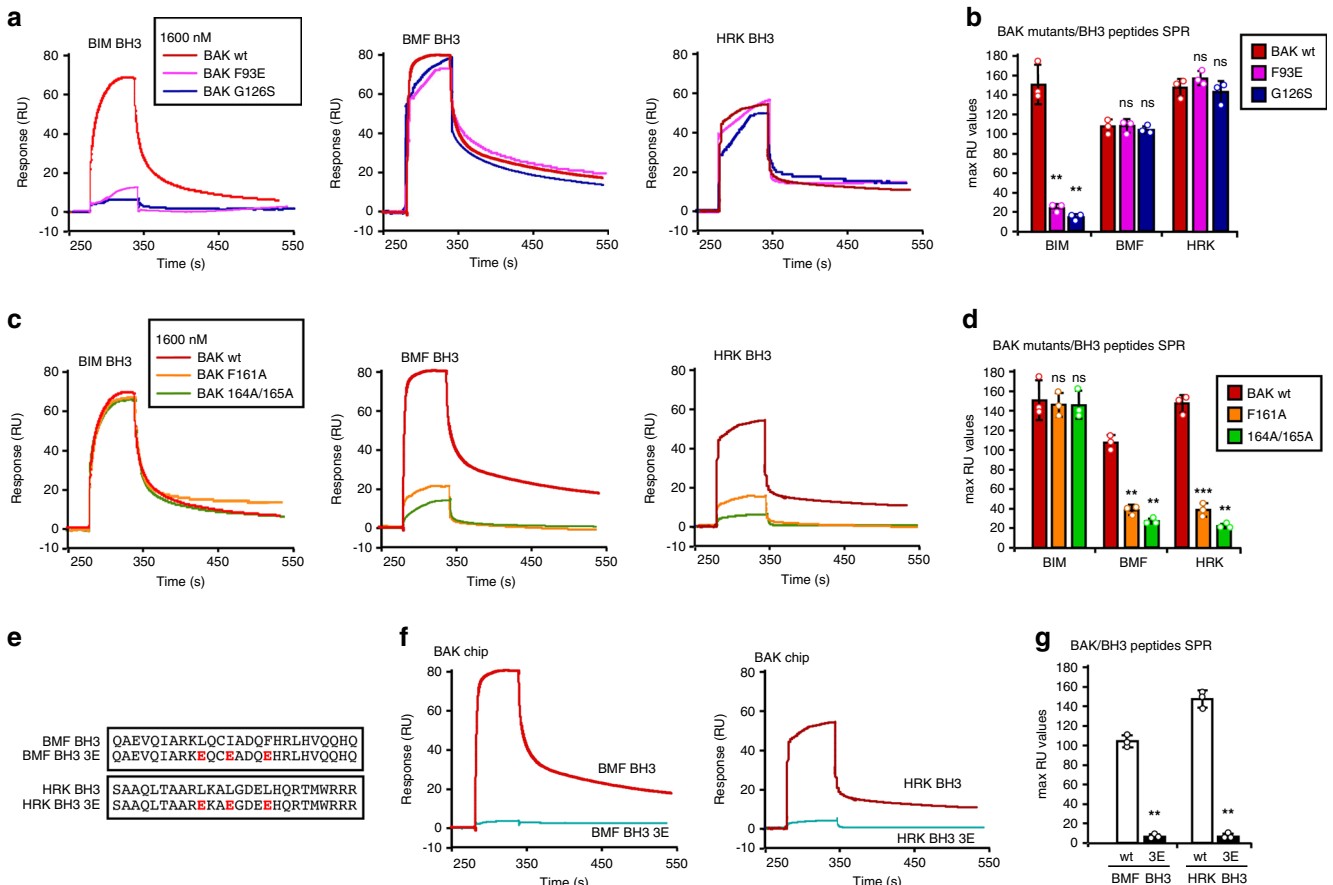

**Fig. 6 BMF and HRK bind the BAK α4/α6/α7 groove. a** SPR (relative units (RU)) as a function of time observed when similar levels (about 4000 RU) of WT BAKΔTM or BAKΔTM with mutations in the canonical BH3-binding groove were immobilized and exposed to 1600 nM BIM BH3 (left), BMF BH3 (middle), or HRK BH3 (right). **b** Maximum RU values obtained from binding isotherms using a series of concentrations of the corresponding peptides in **a**, and $p = 0.007$, 0.008 for BIM BH3; 1.0, 0.48 for BMF BH3; and 0.06, 0.26 for HRK BH3 when maximum RU values of binding to BAK F93E or G126S, respectively, were compared with WT BAK. **c** SPR as a function of time observed when similar levels (~4000 RU) of WT BAKΔTM or the indicated BAKΔTM α4/α6/α7 groove mutant were immobilized and exposed to 1600 nM BIM BH3 (left), BMF BH3 (middle), or HRK BH3 (right). **d** Maximum RU values obtained after exposure of the indicated BAK species to a series of concentrations of the corresponding peptides in **c**. $p = 0.52$, 0.32 for BIM BH3; 0.004, 0.001 for BMF BH3; 0.0003, 0.001 for HRK BH3 when maximum RU values of binding to BAK F161A, or 164/165 A, respectively, were compared to WT BAK. **e** Sequences of 3E BMF and HRK BH3 peptides bearing alterations in the indicated hydrophobic residues. **f** SPR observed when immobilized BAKΔTM was exposed to 1600 nM BMF BH3 or its mutant (left) and HRK BH3 or its mutant (right). **g** Maximum RU values obtained after exposure of BAKΔTM to a series of concentrations of the peptides in **f**. $p = 0.001$ and 0.001 for BMF and HRK BH3, respectively, when maximum RU values observed with the 3E mutants were compared with corresponding WT BH3 peptides. Error bars in **b**, **d**, **g**: mean ± S.D. of three independent experiments. ns, $p > 0.05$; **, $p < 0.01$; ***$p < 0.001$; two-tailed paired $t$ test for maximum RU values of mutants vs WT. Source data are provided as a source data file.

HRK to glutamate (Fig. 6e). These mutations diminished binding of the BH3 peptides to BAKΔTM (Fig. 6f–g), consistent with the MD simulations, suggesting that L137 is involved in the binding at the noncanonical groove and all three hydrophobic residues are involved in binding at the canonical groove[17,25]. Taken altogether, the results shown in Figs. 4–6 suggest that the BAK binding to the BH3 domains of BMF and possibly HRK involves, at least in part, the α4/α6/α7 groove.

**BAK activation is also impacted by α6 helix mutations**. We also examined the impact of the F161A mutation on BAK-mediated liposome permeabilization. As shown in Fig. 7a, this mutation markedly decreased BAK-mediated liposome permeabilization initiated by BMF or HRK BH3 domains, but did not affect liposome permeabilization initiated by BIM or PUMA.

In further studies, we reconstituted $Bak^{-/-}Bax^{-/-}$ DKO MEFs with WT BAK or BAK mutants and examined the cytotoxicity of

transiently expressed BH3-only proteins fused to EGFP (Fig. 7b–d). Apoptosis induction by EGFP-BIM (Supplementary Fig. 8a) or EGFP-PUMA (Fig. 7c) was not inhibited in cells expressing BAK F161A but was inhibited by the BAK G126S mutation even in the presence of the reciprocal N86G mutation (Fig. 7d) that allows BAK oligomerization[17,38]. In contrast, cytotoxicity of EGFP-BMF or EGFP-HRK was markedly diminished by the BAK F161A mutation (Supplementary Fig. 8a and Fig. 7c), reflecting inability of EGFP-BMF or HRK to induce activation of BAK F161A (Fig. 7e), but was relatively unaffected by the BAK G126S/N86G mutation (Fig. 7d and Supplementary Fig. 8b).

To rule out the possibility that decreased ability of BMF and HRK to induce BAK F161A activation results from altered binding of BAK F161A to antiapoptotic BCL2 proteins, we compared the binding of BAK WT and F161A to antiapoptotic BCL2 proteins using pull-down assays. As shown in Fig. 7f, the F161A mutation did not change binding of BAK to BCLX$_L$, BCL2, or MCL1.

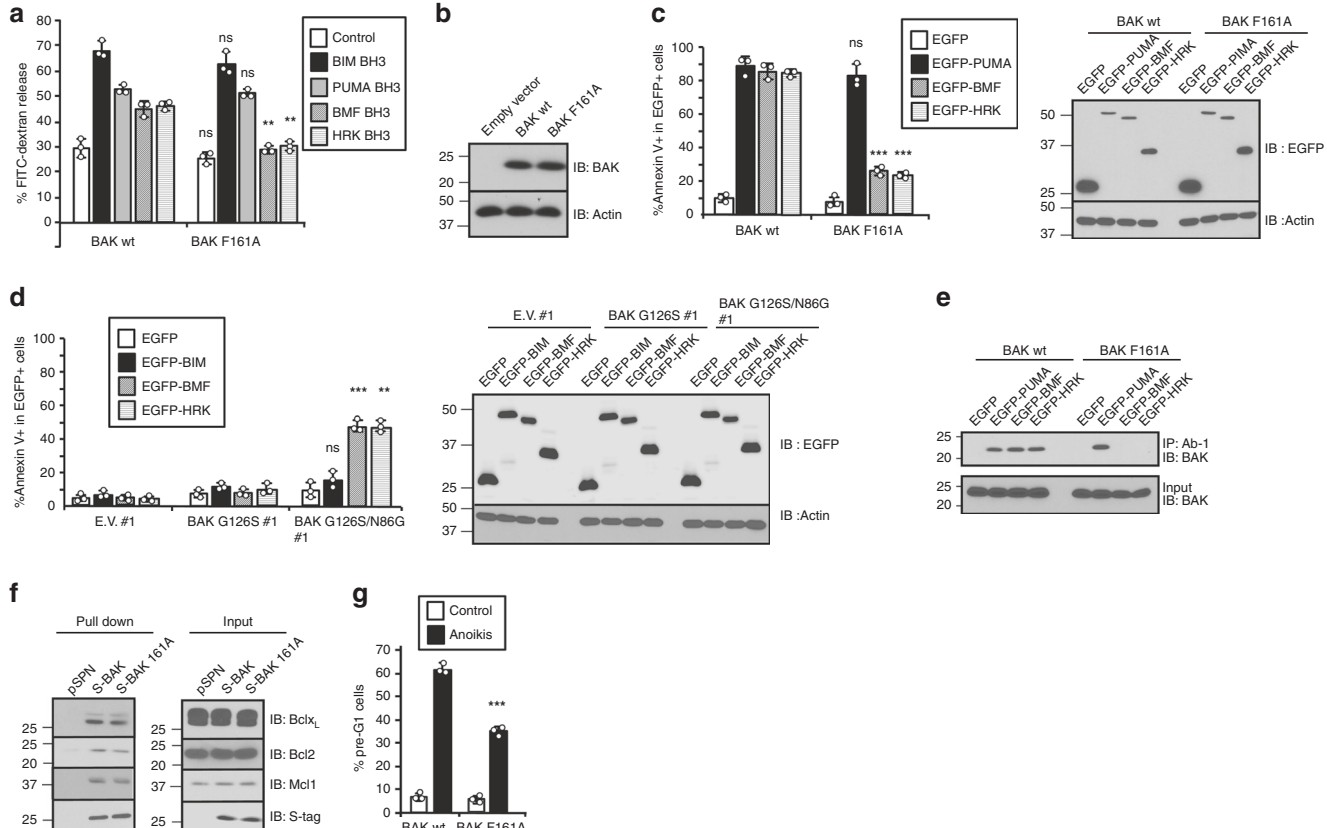

**Fig. 7 Dependence of BMF- and HRK-induced apoptosis on the alternative BAK binding mode. a** Liposome permeabilization assay performed in the presence of 50 nM wild-type (WT) BAKΔTM or BAKΔTM F161A and 50 nM of the indicated BH3 peptides. $p = 0.24, 0.33, 0.20, 0.006$, and $0.003$ for control, BIM, PUMA, BMF, and HRK BH3s, respectively, when liposome release mediated by BAK F161A mutant was compared with WT BAK in the presence of the same BH3 peptide. **b** $Bak^{-/-}Bax^{-/-}$ MEFs were transduced with the indicated retrovirus to express WT BAK and BAK F161A. After 2 weeks of selection, a pool of cells was subjected to western blot. **c–e** After the indicated EGFP-tagged BH3-only proteins were transfected into $Bax^{-/-}Bak^{-/-}$ DKO MEFs reconstituted with WT BAK, F161A BAK **c**, **e** or empty vector (E.V.), BAK G126S without or with the reciprocal N86G mutation that restores BAK oligomerization[38] **d**, cells were harvested, stained with APC-conjugated Annexin V, and subjected to flow cytometry **c**, **d**. The percentages of EGFP+ cells that are Annexin V+ are indicated. $p = 0.18, 0.0006, 0.0002$ for PUMA, BMF, and HRK, respectively, when the percentages of Annexin V+ cells reconstituted with BAK F161A were compared with those reconstituted with WT BAK **c**, and $p = 0.10, 0.0009, 0.001$ for BIM, BMF, and HRK, respectively, when the percentages of Annexin V+ cells of reconstituted with BAK G126S/N86G were compared with those reconstituted with WT BAK **d**. In addition, cell lysates were prepared for immunoprecipitation with BAK Ab-1 antibody and blotting for BAK **e**. **f** After S-tagged WT or F161A BAK was transfected into MEFs for 24 h, cell lysates were subjected to pull-down with S-protein agarose and blotting with the indicated antibodies. **g** Anoikis was induced for 48 h in $Bak^{-/-}Bax^{-/-}$ DKO MEFs reconstituted with WT BAK or BAK F161A. The percentage of cells with sub-G1 DNA was assayed ($p = 0.0006$). Right panels in **c** and **d**, whole-cell lysates subjected to immunoblotting. Error bars: mean ± S.D. of three independent experiments. ns, $p > 0.05$; **$p < 0.01$; ***$p < 0.001$, paired two-tailed $t$ test for comparison of dextran release or cell death induced by mutants vs WT. Source data are provided as a source data file.

To assess BMF-initiated BAK activation under endogenous conditions, we examined detachment-induced apoptosis, a process that is diminished in BMF-depleted cells (Supplementary Fig. 8c)[7]. When this BMF-dependent process was subsequently assessed in $Bak^{-/-}Bax^{-/-}$ DKO MEFs reconstituted with BAK WT or BAK F161A, significantly less anoikis was induced in cells reconstituted with BAK F161A (Fig. 7g), strengthening the view that endogenous BMF might, at least in part, bind the alternative site to trigger BAK-mediated apoptosis.

Collectively, the results in Fig. 7 suggest that canonical BH3-binding groove can be irreversibly occluded by mutations without affecting certain BH3-only proteins such as BMF, but simultaneous disruption of binding to the canonical BH3-binding pocket and α4/α6/α7 groove impairs BAK activation and cell death induced by these proteins.

## Discussion

In the present study, we characterized the interaction of the BH3-only proteins BMF, HRK, and BIK with the mitochondrial permeabilizer BAK. Our experiments not only demonstrated direct interactions between BMF or HRK BH3 and BAK, but also provided evidence that a second site that might play a role in BH3-binding. This second site has potential implications for future efforts to modulate BAK activity.

Our SPR studies indicated that the BAK displays a higher affinity for BMF BH3 than for BIM and PUMA BH3s, two well-established BAK activators. Moreover, BAK also bound the HRK and PUMA BH3s with comparable affinities (Fig. 1). Although BAK showed lower affinity for the BIM and PUMA BH3s than in our previous studies[17,25], the differences might reflect immobilization of BAK rather than BH3 peptides and a different analysis method (Supplementary Fig. 1). Our further experiments showed that the BMF and HRK BH3 domains can induce BAK-mediated liposome permeabilization (Fig. 2a–b) and mitochondrial cytochrome c release (Fig. 2e, g), in agreement with previous studies concluding that BMF and HRK can directly activate BAK[26,27]. Although BIK has also been reported to be a direct BAK activator, in our study interaction of the BIK BH3 domain with BAK was

undetectable using SPR (Fig. 1e). Nonetheless, BIK BH3 peptide at a high concentration (10 μM) induced cytochrome c release from mitochondria of $Bax^{-/-}$ MEFs (Fig. 2e) even though the BAD BH3 peptide, the BCL2/BCLx$_L$ inhibitor navitoclax and/or the MCL1 inhibitor S63845 did not (Fig. 2c–e and Supplementary Fig. 3b), suggesting low but detectable ability of BIK BH3 to activate BAK.

In further experiments, the ability of EGFP-BMF, EGFP-HRK, and EGFP-BIK to induce apoptosis in transfected cells was examined. Previous studies have suggested BAK could be activated through BIM-, PUMA-, and BID-induced direct activation or concentration-dependent autoactivation[15,17,21]. To exclude the influence of these possible mechanisms, we generated HexaKO Jurkat cells, which lack BAK, BAX, BIM, PUMA, BID, and NOXA, introduced an inducible BAK expression construct, and then studied BMF- and HRK-induced apoptosis. Under conditions where BAK is not autoactivated (Fig. 3g) and also in the absence of the well-defined activators BIM, PUMA, BID, and NOXA (Fig. 3e), full-length BMF or HRK can induce BAK-mediated apoptosis (Fig. 3h–i). Moreover, EGFP-tagged full-length BIK also induced more apoptosis than the sensitizer BH3-only protein BAD (Fig. 3h–i), again suggesting that BIK is a weak BAK activator. We note, however, that expression of EGFP-BIK is lower than other BH3-only proteins fused to EGFP, perhaps because of a short BIK half-life in MEFs and Jurkat cells.

Activation of BAK through binding to the canonical BH3-binding groove has been previously demonstrated by NMR[30], x-ray crystallography[29], and site-directed mutagenesis assays[17,25]. In these studies, BH3-only proteins have been shown to interact with the canonical BH3-binding groove of BAK to initiate a BAK conformational change that triggers BAK oligomerization and activation[29,30,38]. Interestingly, the present results suggest that the BMF and HRK BH3 peptides interact, at least in part, with an alternative binding site on BAK. Evidence for this alternative binding site comes from NMR spectroscopy, MD simulations, and site-directed mutagenesis.

In our study, PUMA, BMF, and HRK BH3s all induced the same chemical shift changes of residues in helix α1, including E24, Y41, and Q44, in agreement with previous reports of α1 conformational changes during BAK activation[22,39,40]. Other chemical shift changes, however, differed depending on the BH3 peptide. In particular, the PUMA BH3 peptide bound to the canonical BAK BH3-binding site, as indicated by the chemical shift perturbations or peak disappearance of signals from residues on BAK α3 (G82, D83, I85, Y89, D90, S91, F93, Q94, T95, L97, Q98, H99), α4 (K113, I114, L118), and α5 (G126, A130, L132, G133, F134) (Figs. 4a, 5a and Supplementary Fig. 6b). Some of the signals from residues on α6 and α7 of BAK (such as F157, L163, H165, C166, and A168) were also perturbed or disappeared after adding the PUMA BH3 (Figs. 4a and 5a), possibly because addition of PUMA BH3 caused BAK α4 to move toward α6[30,41]. In contrast, binding of BMF and HRK BH3 domains induced different chemical shift perturbations. Specifically, many of the chemical shift changes or peak disappearances seen with PUMA BH3, including those residues in helix α3 (G82, D83, I85, Y89, D90, S91, F93, Q94, T95, and L97) and helix α5 (G126, A130, L132, G133, and F134), were attenuated or absent when HRK and BMF BH3 peptides bound (Figs. 4b, 5b, c, and Supplementary Fig. 6a, 6c–d) despite the similar or increased affinity of BAK for these BH3 domains (Fig. 1f). Instead, new chemical shift changes involving residues on helix α6 (F161, M162) and helix α7 (A168) appeared when BMF or HRK BH3 was added to BAK. One interpretation of these results is that BMF and HRK BH3 interact less with BAK α3 and α5, i.e., the canonical BH3-binding groove, and more with a site formed by helixes α4, α6 and α7. Although the peak intensity reductions were much reduced in the canonical

groove, our NMR study did show that some of the residues in this groove (e.g., Q94, L97, Y108, L118. G126, L131, and F134) still showed some reduction of peak intensities after BMF or HRK BH3-binding (Supplementary Fig. 6b–d), suggesting that BMF and HRK might possibly bind BAK at two sites.

To further evaluate this possibility, we performed extensive MD simulations of BH3 binding to either the canonical groove or the α4/α6/α7 groove. For each BH3 peptide binding to each site, we obtained 20,000 conformations at 100-ps intervals of the MD simulations and calculated the population of the most populated conformer of each BAK with BH3 binding at either groove. BIM, a BH3 peptide that is well established to bind the canonical groove[17,18,29,41] had a dominant population of 75% at the canonical groove but only 35% at the α4/α6/α7 groove, suggesting the lower affinity of BIM for the alternative groove. In contrast, BMF and HRK BH3 peptides bound to both grooves with populations of 53–74% for both grooves (Supplementary Table 2). These results suggest that (1) BIM binds almost exclusively to the canonical groove, (2) BMF binds to both grooves with a possible slight preference for the noncanonical groove, and (3) HRK binds to both grooves with a preference for the canonical groove.

Further studies examined two types of mutations on binding: those that interfere with binding at only the canonical BH3-binding pocket (G126S, F93E) and those that interfere with binding at both sites by expanding the canonical site and contracting the possible alternative site (F161A and H164A/H165A). Mutations that occlude the canonical BH3-binding pocket[38] abolished the binding of BIM BH3 but not BMF or HRK BH3 to BAK (Fig. 6a–d). In marked contrast, mutations that simultaneously expand the canonical site and narrow the alternative site diminished binding of BMF and HRK BH3s but marginally affected the binding of BIM BH3. These results appear to indicate BMF and HRK BH3s are able to bind to a site on BAK even when the canonical binding groove is occluded[38]. Binding of BMF and HRK to both sites in the wild-type protein and the alternative binding site in the G126S or F93E mutant would certainly be compatible with these results. Consistent with these results, $Bak^{-/-}Bax^{-/-}$ MEFs expressing BAK G126S/N86G were less sensitive to BIM-induced apoptosis than cells expressing WT BAK, but the BAK G126S/N86G mutation had much less effect on BMF- or HRK-induced apoptosis (Fig. 7d). In contrast, the F161A mutation, which affects both grooves, diminished BMF- and HRK-induced apoptosis and the BMF-dependent process of anoikis (Fig. 7g and Supplementary Fig. 8b).

The most straightforward explanation for the NMR, MD simulations, and mutational data is that certain BH3 peptides such as BMF and, to a lesser extent, HRK are able to bind and activate BAK through a previously unrecognized peptide-binding groove comprised of BAK α4, α6, and α7 helices, especially when the canonical BH3-binding groove is irreversibly occluded by mutations. Nonetheless, because the interaction of BMF or HRK BH3 with the α4/α6/α7 groove has not been biochemically confirmed in intact cells, e.g., by cross-linking studies in situ[16], this model remains one of several potential explanations for the results. In particular, the NMR data do not exclude the possibility that the BH3 peptides can bind at the canonical groove and allosterically perturb residues located in the noncanonical groove. However, if BMF or HRK BH3 peptide were binding exclusively to the canonical groove, it is difficult to explain why so many chemical shift perturbations and disappearances seen with PUMA peptide binding at that groove are markedly diminished or absent. Although one potential solution to this conundrum could conceivably be an alternative binding mode in which the BHK and HRK peptides bind in a different orientation across the canonical BH3-binding groove, any model postulating this

different binding mode at the canonical groove must take into account (i) the lack of abundance of any conformer in this alternative orientation in the MD simulations and (ii) the lack of impact of the canonical groove-perturbing G126S and F93E mutations on BMF and HRK binding and action. Nonetheless, further studies such as in situ cross-linking[16] will be required to definitively distinguish between alternative explanations for the results.

The identification of the BAK α4/α6/α7 groove as an alternative BH3-binding site has several possible implications. In particular, this site might serve as a potential target for the development of direct BAK activators to treat disorders where apoptosis is impaired. Conversely, the alternative binding site might serve as a potential receptor for one end of Protac molecules designed for diseases with excessive BAK-induced apoptosis. Either possibility would further validate the alternative binding site and simultaneously provide new insights into BAK function.

## Methods

**Materials**. Reagents were obtained from the following suppliers: lipids and extruder from Avanti Polar Lipids, CM5 biosensor chips from GE Healthcare, Polysorbate 20 from Biacore AB, Q-VD-OPh from SM Biochemicals (Anaheim, CA), glutathione from Sigma, Ni$^{2+}$-NTA-agarose from Novagen, fluorescein iso-thiocyanate (FITC)-labeled dextran 10 from Invitrogen, navitoclax, and S63845 from Chemietek (Indianapolis, IN). Antibodies to the following antigens were purchased from the indicated suppliers and used with the indicated dilutions: Hsp60 (cat. #4870 S, 1:1000), EGFP (#2555 S, 1:1000), BAX (#2772 S, 1:1000), BIM (#2933 S, 1:1000), BID (#2002S, 1:1000), BCLX$_L$(#2764 S, 1:1000), and MCL1 (#4572 S, 1:1000) from Cell Signaling Technology; cytochrome c (#556433, 1:1000) from BD Biosciences; BAK N-term antibody (#06-536, 1:1000) and BAK Ab-1 antibody (TC-100) from Millipore; BCL2 antibody from DAKO (#M0887, 1:1000); NOXA antibody from Enzo Life Sciences (#ALX-804-408-c100, 1:1000); BMF antibody from Proteintech (#18298-1-AP, 1:1000) and actin (goat polyclonal, I-19, #sc-1615, 1:500) and PUMA (#sc-374223, 1:1000) from Santa Cruz Biotechnology. Anti-S peptide antibody was raised in our laboratory and has been validated[42]. BH3 peptides were generated by solid phase synthesis at GL Biochem Ltd. (Shanghai, China).

**Protein expression and purification**. Plasmids encoding BAKΔTM (GenBank BC004431, residues 1–186) in pET29b(+) was kindly provided by Qian Liu and Kalle Gehring (McGill University, Montreal, Canada)[34]. Plasmids encoding BAK 15–186 and the corresponding BAK mutants were generated using site-directed mutagenesis. All plasmids were subjected to automated sequencing to verify the described alterations and confirm that no additional mutations were present.

Plasmids were transformed into E. coli BL21. After these, cells were grown to an optical density of 0.8, isopropyl β- d-1-thiogalactopyranoside was added to a final concentration of 1 mM and incubation was continued for 24 h with shaking at 16 °C. Bacteria were then washed and sonicated intermittently on ice in TS buffer (10 mM Tris-HCl, 150 mM NaCl, pH 7.4, and 1 mM phenylmethylsulfonyl fluoride). His$_6$-tagged proteins were then applied to Ni$^{2+}$-NTA-agarose columns, which were washed with 20 volumes of TS buffer and 10 volumes TS buffer containing 40 mM imidazole, followed by elution with TS buffer containing 200 mM imidazole.

**Surface plasmon resonance**. Proteins for SPR were further purified by FPLC on Superdex S200, concentrated in a centrifugal concentrator (Centricon, Millipore), dialyzed against Biacore buffer (10 mM 4-(2-hydroxyethyl)-1-piper-azineethanesulfonic acid (HEPES; pH 7.4), 150 mM NaCl, 0.05 mM ethylenedia-minetetraacetic acid (EDTA) and 0.005% (w/v) polysorbate 20) and stored at 4 °C for <48 h before use.

BAKΔTM or mutants were immobilized on a CM5 chip using a Biacore T200 biosensor with a blank channel as a negative control. Binding assays were performed at 25 °C using Biacore buffer containing the indicated BH3 peptides injected at 30 μl per min for 1 min using a Biacore T200 Control Software. Proteins were allowed to dissociate during perfusion with Biacore buffer at 30 μl per min for 10 min and then desorbed with 2 M MgCl$_2$. All binding affinities were derived by Biacore T200 Evaluation Software using Steady State analysis (Biacore, Uppsala, Sweden).

**Preparation of FITC-dextran lipid vesicles**. 1-Palmitoyl-2-oleoyl-sn-glycero-3-phosphocholine, 1-plamitoyl-2-oleoyl-sn-glycero-3-phosphoethanolamine, L-α-phosphatidyl-inositol, cardiolipin, cholesterol and 18:1 DGS-NTA(Ni) were mixed at a weight ratio of 36:22:9:8:20:5, and dried as thin films in glass test tubes under nitrogen for 16 h. To encapsulate FITC-labeled dextran 10 (F-d10), 50 mg F-d10 was added to 50 mg lipid in 1 ml of 150 KCl buffer containing 20 mM HEPES

(pH 7.0), sonicated and extruded 15 times through a 400 nm polycarbonate membrane. Liposomes with F-d10 were further purified by gel filtration on Sephacryl S-300 HR (GE Healthcare). Phosphate was determined by colorimetric assay (Abcam, Cambridge, UK).

**Liposome permeabilization assay**. Release of F-d10 from large unilamellar vesicles (LUVs) was monitored by fluorescence dequenching[25] using a fluorimetric plate reader (Thermo Scientific) with SoftMax Pro software. After addition of purified His$_6$-BAKΔTM and/or the BH3 peptides to LUVs (final lipid concentration 10 μg ml$^{-1}$), 96 well plates were incubated at 37 °C and recorded (excitation 485 nm, emission 538 nm) every 10 sec. FITC-labeled dextran release was calculated using the equation (($F_{sample} - F_{blank}$)/($F_{Triton} - F_{blank}$) × 100%), where $F_{sample}$, $F_{blank}$, and $F_{Triton}$ are the fluorescence recorded for the reagent-, buffer-, and 1% Triton-treated LUVs, respectively.

**Immunoblotting**. Cells were solubilized in SDS sample buffer containing 4 M urea, 2% (w/v) SDS, 62.5 mM Tris-HCl (pH 6.8), 1 mM EDTA and 5% (v/v) 2-mer-captoethanol, and sonicated 40 times. Aliquots containing 50 μg of protein were then separated on SDS-PAGE, transferred to nitrocellulose membrane, and probed by antibodies overnight at 4 °C with indicated dilutions, followed by incubation with horseradish peroxidase-labeled secondary antibody at room temperature for 1 h. Membranes were developed using ECL.

**Cytochrome c release**. Mitochondria[43] purified from $Bax^{-/-}$ MEFs were incubated with the indicated concentrations of navitoclax, S63845 or BH3 peptides at 25 °C for 90 min. After centrifugation (10,000 g, 15 min) and washing, supernatants and pellets were analyzed by immunoblotting. Alternatively, His$_6$-BAKΔTM was dialyzed against mitochondria (mito) buffer (150 mM KCl, 5 mM MgCl$_2$, 1 mM EGTA, 25 mM HEPES, pH 7.5) and diluted into mito buffer with 5 mM DTT. Mitochondria purified from $Bak^{-/-}Bax^{-/-}$ MEFs were then incubated with 2 μM His$_6$-BAKΔTM and the indicated BH3 peptides (5 μM) at 25 °C for 90 min. After centrifugation (10,000 × g, 15 min) and washing, supernatants and pellets were analyzed by immunoblotting.

**NMR sample preparation and assignments**. For NMR samples, bacteria were grown in M9 medium. Uniformly labeled BAK 15–186 was produced using M9 medium containing 0.5 g L$^{-1}$ 99% $^{15}$N-ammonium sulfate and 2.5 g L$^{-1}$ 99% $^{13}$C-glucose as the sole nitrogen and carbon sources, respectively. After Ni$^{2+}$-NTA column purification, the labeled protein was further purified using Superdex-75 size exclusion chromatography (GE) and dialyzed against 20 mM NaH$_2$PO$_4$ (pH 6.7) containing 100 mM NaCl, 1 mM DTT, and 2 mM EDTA.

All NMR spectra were recorded at 303 K on a Bruker DMX850 spectrometer with Topspin software. To assign backbone residues of BAK 15–186, a set of standard 2D and 3D spectra was recorded, including $^1$H-$^{15}$N-HSQC (heteronuclear single-quantum correlation), HNCACB, CACB(CO)NH, HNCA, HN(CA)CO. All NMR data were processed with NMRPipe[44] and analyzed with Sparky 3 software[45].

**NMR chemical shift perturbation assay**. In order to define the residues in BAK responsible for the binding of PUMA, BMF, and HRK BH3 peptides, 0.5 mM $^{15}$N-labeled BAK 15–186 was titrated with unlabeled PUMA, BMF, and HRK in different molar ratios. A series of HSQC spectra was then acquired at 303 K on the Bruker DMX850 spectrometer. A plot of chemical shift perturbations (CSPs) across all residues was generated for each BH3 peptide. The threshold value was defined as average CSPs value plus one standard deviation. CSPs for each residues were calculated using the formula CSPs = $[(\Delta\delta NH^2 + \Delta\delta N^2/25)/2]^{1/2}$. The average CSPs values of PUMA, BMF, and HRK peptide were 0.01, 0.02, and 0.01, respectively.

**Cell culture**. Jurkat (T-cell acute lymphoblastic leukemia) cells (obtained from Paul Leibson, Mayo Clinic) were maintained below $10^6$ cells ml$^{-1}$ in RPMI 1640 medium containing 10% heat-inactivated fetal bovine serum, 100 units ml$^{-1}$ penicillin G, 100 μg ml$^{-1}$ streptomycin, and 2 mM glutamine. MEFs (WT, $Bak^{-/-}$, $Bax^{-/-}$, $Bak^{-/-}Bax^{-/-}$, obtained from Andrew Badley, Mayo Clinic) were maintained in Dulbecco's modified Eagle's medium (DMEM) medium containing 10% heat-inactivated fetal bovine serum, 100 units ml$^{-1}$ penicillin G, 100 μg ml$^{-1}$ streptomycin, and 2 mM glutamine.

**Mammalian expression plasmids and transfection**. cDNAs encoding human BIM$_{EL}$ (GenBank AF032457), PUMA (GenBank NM_001127242), NOXA (Gen-Bank NM_021127), BAD (GenBank AF021792), BMF (NM_001003940.2), and HRK (NM_003806.4) were cloned into XhoI and BamHI sites of pEGFP-C1 to yield constructs fused at the C termini of EGFP[46]. Empty pEGFP-C1 served as a control. All plasmids were verified by Sanger sequencing.

Log phase Jurkat cells or MEFs growing in antibiotic free medium were transiently transfected with the indicated plasmid using a BTX 830 square wave electroporator delivering a single pulse (10 msec) at 240 mV or 280 mV for Jurkat or MEFs, respectively. Cells were incubated for 24 h, stained with APC-coupled annexin V, subjected to flow microfluorimetry, and analyzed using CytExpert software[47].

**HexaKO Jurkat cells and inducible BAK expression.** Human BAK, BIM, PUMA, BID, NOXA, and BAX were sequentially knocked out using CRISPR-Cas9 technique in Jurkat cells[46]. The targeting sequences were as follows: BAK: 5′-ACGG CAGCTCGCCATCATCG-3′; BIM: 5′-GCCCAAGAGTTGCGGCGTAT-3′; PUMA: 5′-CAGCTCCCCGGAGCCCGTAG-3′; BID: 5′-CGCAGAGAGCTGG ACGC ACT-3′; NOXA: 5′-CGCTCAACCGAGCCCCGGCGC-3′; and BAX: 5′- CG AGTGTC TCAAGCGCATC-3′. In brief, individual oligonucleotides were synthesized, annealed, and cloned into the BsmBI site of lentiCRISPR-V2 (Addgene) plasmid. Viruses were packaged in HEK293T cells (obtained from Richard Bram, Mayo Clinic) by transfecting using Lipofectamine 2000 with the packaging vector psPAX3, envelope vector pMD2.G, and lentiCRISPR-v2 encoding a targeting sequence to the individual molecule mentioned above. After transducing cells with individual viruses sequentially, Jurkat cells were selected with 3 μg ml$^{-1}$ puromycin and cloned by limited dilution. The knockouts were verified by immunoblotting.

To allow controlled BAK expression in HexaKO Jurkat cells, we created a stable cell line in which BAK expression can be induced by doxycycline. In brief, a CRISPR/Cas9-resistant version of BAK (in which the CRISPR/Cas9 targeted sequence of BAK was mutated to 5′-ACGTCAGCTGGCCATAATTG-3′ and no amino acids were changed) was generated by site-directed mutagenesis in pcDNA3.1 using wild-type BAK plasmid as a template[21]. The knockout-resistant BAK mutant was PCR amplified and cloned into the virus backbone of pRRL. TRE3G.Blast.IRES.rtTA3 plasmid (kind gift from Dr. S. Yun, Moffitt Cancer Center) using BamHI and EcoRI sites. Viruses were then packaged in HEK293T cells by transfecting this plasmid with the packaging vector psPAX3, envelope vector pMD2.G using lipofectamine 2000 and transduced to Jurkat HexaKO cells. After 2 days, cells were selected with 5 μg ml$^{-1}$ blastocidin. Following cloning by limiting dilution, two stable clones were confirmed by western blotting under inducible conditions.

BAK was induced in the selected clones by treatment with doxycycline at the indicated concentrations for 16 h. When EGFP-BH3-only plasmids were transfected into these cells, BAK was induced using 10 ng mL$^{-1}$ doxycycline for 16 h and cDNAs encoding the indicated EGFP-BH3-only proteins were then transfected by electroporation.

**Stable cell lines.** $Bak^{-/-}Bax^{-/-}$ MEFs were maintained in DMEM supplemented with 10% FCS, 250 μM L-asparagine and 55 μM 2-mercaptoethanol. 293 T cells were maintained in DMEM supplemented with 10% FCS. WT BAK or mutants were stably expressed in DKO MEFs by retroviral transduction. 24 h after transduction of DKO MEFs with WT BAK or mutant BAK in pBabe-puro(+), cells were selected in DMEM (10% FCS) with 2 μg ml$^{-1}$ puromycin for 2 weeks. Pooled cells were examined by western blotting and used for the indicated experiments.

**BAK Ab-1 antibody immunoprecipitation.** After MEFs were transfected with EGFP-tagged BH3-only proteins, cells were incubated in presence of 10 μM Q-VD-OPh for 24 h and harvested for lysis in CHAPS buffer (1% CHAPS, 1% glycerol, 150 mM NaCl, and 20 mM HEPES, pH 7.4). Equal amounts of lysates (400 μg) were then incubated for 24 h at 4 °C with BAK Ab-1 antibody that was precoupled to protein A/G agrose beads using dimethyl pimelimidate[17]. After four washes with isotonic wash buffer containing 1% CHAPS, bound proteins were solubilized in SDS sample buffer, subjected to SDS-PAGE and probed with antibodies that recognize total BAK.

**Immunoprecipitation of antiapoptotic BCL2 proteins.** After BAK was induced for the indicated time in presence of 10 μM Q-VD-OPh, cells were harvested for lysis in CHAPS buffer. Equal amounts of lysates (400 μg) were then incubated for 24 h at 4 °C with BCL2, MCL1, or BCLX$_L$ antibody that was precoupled to protein A/G agarose beads using dimethyl pimelimidate[21]. After four washes, bound proteins were solubilized in SDS sample buffer, subjected to SDS-PAGE and probed with the indicated antibodies.

**Pull-down assay.** After S-tagged BAK or BAK mutant was transfected into $Bak^{-/-}Bax^{-/-}$ MEFs in presence of 10 μM Q-VD-OPh for 24 h, cells were harvested for lysis in CHAPS buffer. Equal amounts of lysates (400 μg) were then incubated for 24 h at 4 °C with S-protein agarose beads. After four washes, bound proteins were solubilized in SDS sample buffer, subjected to SDS-PAGE, transferred to solid support and probed for the indicated proteins.

**Anoikis assay.** Anoikis was assayed using the poly-HEMA method. In brief, poly-HEMA (Sigma, Shanghai, China) was dissolved in ethanol for stock (120 mg ml$^{-1}$). The stock solution was then diluted 10-fold with ethanol and transferred into 12-well plates (50 μL per well). The plates were then incubated at room temperature for 48 h until dry. The wells were washed three times with PBS solution and then one time with DMEM solution before use. The $Bax^{-/-}$ MEFs were transfected with BMF siRNAs, and cultured for 24 h. To induce anoikis, the cells were trypsinized and seeded on the poly-HEMA coated plates. After incubation for 48 h, the cells were collected and prepared for apoptosis detection.

**CD spectroscopy.** BH3 peptides or BAK as a positive control at 0.2 mg ml$^{-1}$ were dissolved in HS buffer (20 mM HEPES, 150 mM NaCl, pH 7.0). CD spectra were collected at room temperature on an AVIV 62DS model spectropolarimeter with a 0.1 cm cuvette. Two sequential scans from 190 nM to 250 nM were recorded and the background spectrum of the buffer only was subtracted. The CD data were then uploaded to k2d2 algorithm (http://coot.embl.de/~andrade/k2d2/k2d2.pl) for α helicity calculation.

**MD simulations.** To simulate HRK BH3 peptide binding at the noncanonical groove, the BH3 peptide of HRK in the α-helical conformation was manually docked onto the α4/α6/α7 groove of BAK (residues 21–183) taken from the crystal structure of a Bak homodimer[34]. This placed the peptide atop Y108$^{BAK}$ and H165$^{BAK}$, L10$^{HRK}$ close to Y108$^{BAK}$ and T112$^{BAK}$, and H18$^{HRK}$ close to F119$^{BAK}$ and F161$^{BAK}$. Topology and coordinate files of the peptide-bound BAK complex were generated by the tLeaP module of AmberTools 13 (University of California, San Francisco). The energy minimization and MD simulations were performed using SANDER and PMEMD of AMBER 16 (University of California, San Francisco) with forcefield FF12MC[35]. The energy minimization used dielectric constant of 1.0, and 100 cycles of steepest-descent minimization followed by 900 cycles of conjugate-gradient minimization. The energy-minimized complex was then solvated with TIP3P water molecules[48] using Solvatebox M TIP3PBOX 8.2 and energy-minimized for 100 cycles as above followed by 900 cycles of conjugate-gradient minimization to remove close van der Waals contacts. The resulting system was then heated from 5 to 340 K at a rate of 10 K ps$^{-1}$ under constant temperature and volume, equilibrated for $10^6$ timesteps under constant temperature of 340 K and constant pressure of 1 atm employing isotropic molecule-based scaling, and finally simulated in 20 31.6-ns, distinct, independent, unrestricted, unbiased, and isobaric–isothermal MD simulations using a periodic boundary condition at 340 K and 1 atm with isotropic molecule-based scaling. The 20 unique seed numbers for initial velocities of Simulations 1–20 were taken from ref. [49]. All simulations used (i) a dielectric constant of 1.0, (ii) Langevin thermostat[50] with a collision frequency of 2 ps$^{-1}$, (iii) the Particle Mesh Ewald method to calculate electrostatic interactions of two atoms at a separation of >8 Å[51], (iv) Δt = 1.00 fs of the standard-mass time[35,52], (v) SHAKE-bond-length constraints applied to all the bonds involving the H atom, (vi) a protocol to save the image closest to the middle of the primary box to the restart and trajectory files, (vii) the revised alkali and halide ions parameters[53], (viii) a nonbonded cutoff of 8.0 Å, (ix) the atomic masses of the entire simulation system (both solute and solvent) that were reduced uniformly by tenfold[35,52], and (x) default values of all other inputs of the PMEMD module. The FF12MC parameters are available in the Supporting Information of ref. [52]. The complex conformations were saved at 100-ps intervals in all simulations. Conformational cluster analysis of the first-round MD simulations of the BAK in complex with the BH3 peptide of HRK identified a complex conformation from Simulation 4 that was consistent with the experimentally determined mutation data. This conformation was then used as the initial conformation for the second-round MD simulations under the same simulation conditions except that (i) the temperature was changed to 303 K that was used in the NMR study and (ii) each of the 20 simulations was increased to 316 nanoseconds. The cluster analysis of all saved complex conformations was performed using the CPPTRAJ module of AmberTools 16 with the average-linkage algorithm[54] (epsilon = 2.0 Å; RMS on the CA atoms of residues 4–23 of the BH3 peptide and Residues 102–125 and 147–177 of BAK). The most populated conformation of the second-round MD simulations identified by the cluster analysis was used as the computational model of the BAK in complex with the BH3 peptide of HRK shown in Fig. 5. The second-round MD simulations were then repeated with the F161A BAK mutant.

To simulate BMF and BIM BH3 binding at the BAK noncanonical groove, the second-round MD simulations of the HRK BH3 peptide at the noncanonical groove were then repeated with the BAK complex with the BH3 peptide of BMF or BIM that was mutated from the initial conformation of the second-round MD simulations of the BAK in complex with the BH3 peptide of HRK. The BMF simulations were then repeated with the F161A BAK mutant.

To simulate BIM, BMF, and HRK BH3 binding at the canonical BAK groove, a BH3 peptide in the α-helical conformation was manually docked onto the α3/α4/α5 groove of the same BAK as the one for HRK binding at the noncanonical groove. The peptide was placed in such a way that its conserved Leu residue was as close to I114$^{BAK}$ and L118$^{BAK}$ as possible. The resulting complex was then refined with energy minimization and simulated under the same simulation conditions as the ones for HRK binding at the noncanonical groove except that (i) the temperature was changed to 303 K and (ii) each of the 20 simulations was increased to 316 ns. These simulations were repeated with F161A BAK mutant. All MD simulations were performed on 100 dedicated 12-core Apple Mac Pros with Intel Westmere (2.93 GHz) processors. The autonomous folding data of CLN025 listed in Supplementary Table 2 were taken from ref. [35].

**Statistical analysis and reproducibility.** The p values described in this study were calculated by paired two-tailed student's t test using Microsoft Excel 2007. Experiments in Figs. 2c–g; 3c, e, f, g; 7b, e–f; Supplementary Fig. 3a–c; and Supplementary Fig. 5a have been repeated twice and similar results were observed.

**Reporting summary**. Further information on research design is available in the Nature Research Reporting Summary linked to this article.

## Data availability
The data supporting the findings of this study are available in the manuscript and supplementary files or are available from the corresponding authors upon request. Source data are provided with this paper.

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

## Acknowledgements

This work is supported by the National Natural Science Foundation of China (no. 81572948, no. 21772201, no. 21703254), the Hundred-Talents Program of Chinese Academy of Science to Haiming Dai, the innovative program of Development Foundation of Hefei Center for Physical Science and Technology (2018CXFX007), Hefei Foreign Cooperation Project (ZR201801020002), the Grant of the President Foundation of Hefei Institutes of Physical Science of Chinese Academy of Sciences (YZJJ201623). We thank the High Magnetic Field Laboratory of Chinese Academy of Sciences at Hefei for recording the NMR spectra and Dr. S. Yun from the Moffitt Cancer Center for the doxycycline inducible plasmid.

## Author contributions

H.D. and S.H.K. conceived and designed the study. K.Y, W.X.M., H.S., B.W., M.C., J.G., H.W., J.W., and H.D. performed the experiments. Y-P.P. designed, performed, and analyzed the MD simulation study. H.D. and S.H.K. wrote the paper. All authors reviewed the manuscript.

## Competing interests

The authors have no conflict of interest related to this research.
