## [Peer Review File · Nature Communications]

Reviewers' comments:

Reviewer #1 (Remarks to the Author):

This is a very interesting article in which the authors propose that two poorly characterized BH3-only proteins, HRK and BMF, bind to Bak at an interface previously not reported, composed of the alpha4, alpha6 and alpha7 helices. I am very intrigued by some of the mutagenesis data to support this, but have concerns about some technical aspects of the paper. I also think there are alternative explanations of the data that the authors should acknowledge as possibilities.

The main concern on alternative possibilities is that BMF and HRK are actually binding to the canonical groove but that this binding pushes alpha 4 over to alpha 6 and 7 more than Bim or other activators. It has been shown in structural studies that alpha 4 moves towards alpha 6 (Moldoveanu, Nat Struc Mol Biol, 2013; Brouwer, Mol Cell, 2017), so is the effect the authors are seeing due to BMF and HRK requiring alpha 4 to shift further than with PUMA or BIM? That would explain chemical shifts concentrated in the region, but also still being seen in alpha 3, eg Q98 and H99 shown in Fig 4b and 4c. The mutation might somehow disrupt this interaction. Alternatively, perhaps binding of BMF and HRK need the residues 161 or 164/165 to support the canonical groove in some way that Bim doesn't.

Unfortunately, it seems to me that there are no simple experiments to prove these possibilities vs the proposed model conclusively. An experiment that would prove the model definitively (besides a structure of the complex) would be to use a peptide with a crosslinker (eg as used by Leshchinera, PNAS, 2013) and to show the cross link occurred in the a4/a6/a7 groove rather than the canonical groove, although that is not a trivial experiment. As such, I would recommend caution on behalf of the authors and to include in their discussion the possibilities described above. It is also important to raise these possibilities for readers not so familiar with the field.

Also, there isn't much of a groove at the new site, and it is actually colluded by F161 and H165. In some ways it is surprising that mutations to Ala don't make it more receptive to peptides if they do bind there. Could the authors comment on this?

Also, it is somewhat surprising that the conserved BH3 motif could target so many different sites on the same protein, this seems to be very unusual for protein-protein interactions. Is there anything about the HRK and BIK BH3 domains would target them to this unique site rather than the canonical domain?

Additionally I have some technical concerns as indicated below:

Some of the SPR curves look like they are prone to artefact. For example:

- Fig 1B. Why are the 1600 nM and 2500 nM curves anomalous? (i.e. rise to a peak then drop then rise again)?

- Fig 1D. Why is there a different on-rate for the 100 to 800 nM curves compared to 1600 and 2400? Presumably one of these is effected by bulk shift. This will impact the ability to assign affinities based on this data.

Fig 1F and Text. How are the K_d 's calculated? Steady State or Kinetic analysis? To my eye kinetic analysis is not appropriate for these curves, certainly fitted curves have not been included so I presume the authors have not performed kinetic analysis. If Steady State then curves for steady state vs concentration should be provided to assess whether these are linear or not (i.e appropriate for affinity analysis). Also, which curves were used in analysis. This is important given that some curves don't reach equilibrium (therefore can't be used for steady state), some curves show artefacts for the top concentrations, and for Bim there are different shaped curves for different concentrations indicating an artefact.

Given that many of the curves appear to be effected by artefact effects the authors should explain these experiments carefully. For example, what was used as a negative for these binding curves, importantly, how do these avoid effects from bulk shift.

Sup Fig 1 – the authors claim “Circular dichroism spectroscopy revealed similar degrees of α -helical-forming abilities of these BH3 peptides”. I am very surprised by the values presented in Sup Fig 1C based on the curves. BIK and HRK are clearly different to BMF in Supp fig 1b. Also, Bik and HRK do not look like classical CD spectra from samples predominately alpha helical.

Fig 2 C-F. Obatoclax has been shown to kill cells independent of Bax and Bak (Vogler et al., Cell Death and Diff, 2009), and its activity on pro-survival proteins is controversial. To inhibit the spectrum of pro-survivals a better combination would be navitoclax with one of the Mcl-1 specific inhibitors published by Abbvie or Servier.

Fig 3A and B – It is difficult to draw conclusions regarding Bik in these cells as the expression is very low.

Fig 3E – Bik should be included in this analysis.

Fig 3G and F – I don't understand how the values for Annexin V positive cells are obtained. Eg. For Bik, it is listed as 51.1%, but the upper quadrants are 18.8% and 10.1% - 28.9%. Same is true for other FACS plots.

Fig 6B, D and G. Again, how were these affinities measured, steady state vs kinetic analysis, and based on the points raised above, is that analysis appropriate for these types of curves. In any case, the effect seems pretty clear cut. It could be represented differently to affinity, eg by max RU values, unless of course those RUs are effected by bulk shift or other artefact as discussed above.

All NMR plots – Graphs of chemical shift perturbations should be plotted as a function of BAK residue with the significance threshold used in fig 5 shown. This is standard for such analysis.

Reviewer #2 (Remarks to the Author):

Bax and Bak are two pore-forming proteins that can induce mitochondrial outer membrane permeabilization (MOMP) and apoptotic cell death. This activity is hence tightly regulated, mainly by the BH3 proteins that activate them and the antiapoptotic Bcl-2 family proteins that inhibit them. Unlike Bax that is a cytosolic protein in healthy cells, Bak is an integral membrane protein in the mitochondrial outer membrane (MOM). This implies that these two proteins might be activated via different mechanisms. One possibility is that different BH3 proteins are employed to activate Bax or Bak. Another possibility, not mutually exclusive from the first one, is that Bax and Bak are activated via different binding sites for BH3 proteins. Indeed, a binding site unique to Bax was found to be a binding site for BH3 proteins such as Bim and tBid to trigger Bax activation in the cytosol, which thereby was termed the trigger site. This trigger site, composed of the helices 1 and 6, is not the binding site for BH3 proteins Bim and tBid to activate Bak. Instead, a canonical BH3-binding site shared by Bak and Bax, also by all the antiapoptotic Bcl-2 proteins, is used for BH3 proteins Bim and tBid as well as Puma and Noxa to activate Bak in the MOM. Since this canonical BH3-binding site is blocked by the helix 9 in the cytosolic Bax protein, it would not be bound by a BH3 activator protein. However, once the soluble Bax protein inserts into the MOM using the helix 9, the canonical binding

site would be free to accept a BH3 protein. According to another model, the binding of a BH3 protein to the canonical site is required for the Bax activation, like the Bak activation.

Now the Dai and Kaufmann groups provide evidence for a novel binding site in Bak for other BH3 proteins BMF and HRK to activate Bak. The evidence includes the SPR detected binding of the BMF and HRK BH3 peptides to Bak protein, the liposomal and mitochondrial membrane permeabilization induced by the BH3 peptides-activated Bak protein, and the apoptotic cell death induced by overexpression of BMF and HRK proteins that depended on Bak. Most importantly, the NMR based chemical shift perturbation data indicated a binding site composed of helices 4, 6 and 7 for the BMF and HRK BH3 peptides, which is different from the binding site indicated by the control BH3 peptide from Puma that binds to the canonical site. Guided by the structural model, mutations in the helix 6 in the novel binding site reduced the binding of BMF and HRK, but not Bim, BH3 peptides to Bak protein. Moreover, mutations in the BMF and HRK BH3 peptides reduced the binding to Bak protein. These mutations, when introduced to the Bak or BMF and HRK proteins that were overexpressed in cells, inhibited apoptosis.

The conclusions made by the authors are well supported by solid experimental evidence. The discovery of the novel binding interaction for the two understudied BH3 proteins is significant not only to the basic understanding of how Bak is activated to induce cell death but to the therapeutic targeting of Bak to kill cancer cells. However, the following issues need to be addressed before the acceptance of the manuscript for publication.

1. The activity of BMF and HRK proteins, as well as Bak, was only determined in the cells that overexpress them or their mutants. Can the authors repeat this assay in the cells whose apoptosis is dependent on the activation of endogenous BMF or HRK and Bak to determine whether the novel interaction is critical?
2. The K_d values for HRK and BIK BH3 peptides binding to Bak protein are different by > 10-fold (Fig 1f), yet, the mitochondrial Cyto C release by these peptides is the same (Fig 2e). What is the explanation?
3. What are the concentrations of the BH3 peptides and Bak protein used in Fig 2f? In this figure, the Cyto C release activities of HRK and BIK BH3 peptides in the mitochondrial containing full-length Bak protein are different. In comparison, their activities in the mitochondria supplemented with a truncated Bak protein are the same. Why?
4. In line 173-174, "BIK...induced cell apoptosis in more than 75% of EGFP+ cells..."; yet in Fig 3a, the percentage is ~40.
5. Fig 3a, the right panel shows that BMF and BIK proteins were expressed at lower levels than other BH3 proteins. Thus, the apoptotic activities shown in the left panel need to be normalized by the

protein levels in the cells. Similar issues with these or other proteins in Fig 3b to d, g, and h also need to be resolved.

6. In Fig 5a, the perturbation of Bak structure by Puma BH3 peptide spreads into the novel site that was perturbed by the BMF or HRK peptide. The authors used the Bak structure bound by the Bid peptide to explain the cause without invoking the possibility that the Puma peptide may also bind the novel site. This is not convincing because the Puma peptide is different from the Bid peptide. Then in the functional assays shown in Fig 6 and 7, they used the Bim peptide and protein as the control, but not the Puma peptide and protein which they should use to validate the structural models. In particular, does the mutations in the novel site reduce the Puma to Bak, and the Puma induced membrane permeabilization and cell death by Bak?

7. All the mutations made for the novel site are in the helix 6. Since the site also composed of other helices, they should make other mutations, in particular, helix 4, to further confirm the structural model.

8. To validate the cell assay data in Fig 7c, they need to show the expression levels of Bak wt and mutant protein.

Reviewed by Jialing Lin

Reviewer #3 (Remarks to the Author):

Ye et al. present a biochemical and structural investigation of BMF and HRK BH3-only proteins and pro-apoptotic BAK to demonstrate that BMF and HRK can directly activate BAX in vitro and in cells using an alternative binding site of BAK. This site is formed by $\alpha 4/\alpha 6/\alpha 7$ helices instead of the canonical c-terminal groove that previous activators such as BID and BIM BH3 domains were found to bind. This is an interesting study in terms of the proposed alternative binding site of BAK, however BMF and HRK have been previously also shown that directly activate BAK. The authors need to provide more convincing case for the proposed binding site and direct activation of BAK.

The NMR data are not so clear. They need to be quantified across all residues and provide chemical shift perturbations plots for each peptide to appreciate significant chemical shifts changes in all residues of BAK and determine the binding site. It is not clear if all residues of the protein have been assigned which may be critical to the definition of the binding sites. Supplemental data in tables for assignments and residues undergoing chemical shift perturbations will be also helpful.

In addition, the alternative pocket is poorly defined by the presented data and the images in figure 5. Only a few residues are highlighted by NMR, how these residues comprise a binding site for these peptides? what are the residues of the groove and how the peptides can be predicted to interact with this proposed site? Is it feasible in terms of the interaction of each binding partner?

The experiments in figures 2 and 3(except the liposomal assay, see below comment) do not discriminate between direct and indirect activation of apoptosis through BAK or BAX and this is not discussed in the text. Similar data have been also described in other publications regarding the apoptotic activity of overexpressed BH3-only proteins. I do not see any experimental condition that suggests that these proteins activate directly BAK in cells.

The experiments in figure 7 show the potential of this novel region to regulate binding and activation of BAK-mediated pore activity and apoptosis. Authors should also test the mutants presented in the binding studies for the new site and the canonical BH3 domain and confirm how the canonical site mutants impair the activation. The should also provide biochemical data from the cell-based experiments that BAK is activated/oligomerized by BMF and HRK proteins from the specific interaction with the proposed site.

Other comments:

The liposomal release mediated by BAK alone is high ~30%, do the authors use conditions and liposome composition that promote BAK auto-activation and BH3-mediated BAX activation?

In figure 3g, BIK is not expressed however there is significant apoptotic activity, how the authors can explain this discrepancy? Moreover, in figure 3C and 3D again there is very low expression of BIK proteins compared to other proteins but still the effect on Annexin V is significant and comparable to other BH3-only proteins.

It is surprising that Navitoclax and Obatoclax at these high doses are not able to induce cytochrome c release. Have the authors used a novel selective MCL-1 inhibitor such as S63845 or AM176 and its combination with Navitolcax?

Reviewer #1 (Remarks to the Author):

This is a very interesting article in which the authors propose that two poorly characterized BH3-only proteins, HRK and BMF, bind to Bak at an interface previously not reported, composed of the $\alpha 4$, $\alpha 6$ and $\alpha 7$ helices. I am very intrigued by some of the mutagenesis data to support this, but have concerns about some technical aspects of the paper. I also think there are alternative explanations of the data that the authors should acknowledge as possibilities.

The main concern on alternative possibilities is that BMF and HRK are actually binding to the canonical groove but that this binding pushes $\alpha 4$ over to $\alpha 6$ and $\alpha 7$ more than Bim or other activators. It has been shown in structural studies that $\alpha 4$ moves towards $\alpha 6$ (Moldoveanu, Nat Struct Mol Biol, 2013; Brouwer, Mol Cell, 2017), so is the effect the authors are seeing due to BMF and HRK requiring $\alpha 4$ to shift further than with PUMA or BIM? That would explain chemical shifts concentrated in the region, but also still being seen in $\alpha 3$, eg Q98 and H99 shown in Fig 4b and 4c. The mutation might somehow disrupt this interaction. Alternatively, perhaps binding of BMF and HRK need the residues 161 or 164/165 to support the canonical groove in some way that Bim doesn't.

Unfortunately, it seems to me that there are no simple experiments to prove these possibilities vs the proposed model conclusively. An experiment that would prove the model definitively (besides a structure of the complex) would be to use a peptide with a crosslinker (eg as used by Leshchinera, PNAS, 2013) and to show the cross link occurred in the $\alpha 4/\alpha 6/\alpha 7$ groove rather than the canonical groove, although that is not a trivial experiment. As such, I would recommend caution on behalf of the authors and to include in their discussion the possibilities described above. It is also important to raise these possibilities for readers not so familiar with the field.

Response: We thank for the Reviewer for the very positive comment on our mutagenesis data that support the binding of the BH3 peptides of BMF and HRK to a previously unrecognized site and the suggestion of evaluating alternative possibilities to confirm the new site binding. We subsequently performed molecular dynamics simulations and found that the two peptides can theoretically bind both the canonical and the $\alpha 4/\alpha 6/\alpha 7$ grooves but their predicted affinities for the latter are much higher than the former according to the simulations. In the revised manuscript, we have added these simulation data to provide further support that the BMF BH3 and HRK BH3 domains bind to the $\alpha 4/\alpha 6/\alpha 7$ groove of BAK (Table S2 and Figs. 5d-5h). As suggested by the Reviewer, however, we have also noted in the discussion the possibility that the two peptides can bind to the canonical groove under certain conditions and that crystallographic and/or crosslinker studies would be needed to provide additional support for the binding to the $\alpha 4/\alpha 6/\alpha 7$ groove (paragraph 2, p. 19).

Also, there isn't much of a groove at the new site, and it is actually colluded by F161 and H165. In some ways it is surprising that mutations to Ala don't make it more receptive to peptides if they do bind there. Could the authors comment on this?

Response: As described in our revised manuscript (p. 11, last paragraph), molecular dynamics simulations showed that the two occluding residues, Y108 and H165, change their side-chain conformations to expose the $\alpha 4/\alpha 6/\alpha 7$ groove upon binding of the BH3 peptides of BMF and HRK. These changes in side-chain conformations are reminiscent of the conformational changes of R88 and Y89 at the canonical groove initially observed in our molecular dynamics simulations of BAK in complex with the BIM BH3 peptide and later validated by crystallography (Dai H., *J. Cell Biol.* 2011, 194: 39-48; Moldoveanu T., *Nat. Struct. Mol. Biol.*, 2013, 20: 589-597).

Our simulations also revealed that F161 remained in the groove to interact with conserved L137 of BMF or the conserved L37 of HRK. It is therefore conceivable that the F161A mutation would disable the interaction between F161 of BAK and L137 of BMF (or L37 of HRK) because the side chain of Ala is not long enough to reach the side chain of L137 or L37.

Also, it is somewhat surprising that the conserved BH3 motif could target so many different sites on the same protein, this seems to be very unusual for protein-protein interactions. Is there anything about the HRK and BIK BH3 domains would target them to this unique site rather than the canonical domain?

Response: This is an interesting question. The BH3 domain is not very conserved because this domain is defined by the presence of only two conserved residues over a 15 residue stretch (Fig. 1a). With this in mind, it might be important that the BMF and HRK BH3 peptides are more positively charged than the BIM BH3 peptide. This difference may explain why the first two bind at the $\alpha 4/\alpha 6/\alpha 7$ groove that abuts a more negatively charged region and why the last one binds at the canonical groove that abuts a less negatively charged region.

Additionally I have some technical concerns as indicated below:

Some of the SPR curves look like they are prone to artefact. For example:

- Fig 1B. Why are the 1600 nM and 2500 nM curves anomalous? (i.e. rise to a peak then drop then rise again)?

Response: The peak drop and rise again in BAK interaction with BMF BH3 peptide might result from small differences between the dilution buffer of BMF BH3 peptide and the running buffer. This might also explain why the phenomenon was only observed at high concentrations. Because the curve drop was observed at 2400 nM BMF BH3, this concentration was not included in the Steady State analysis (shown as Fig. S1a).

- Fig 1D. Why is there a different on-rate for the 100 to 800 nM curves compared to 1600 and 2400? Presumably one of these is effected by bulk shift. This will impact the ability to assign affinities based on this data.

Response: This might be due to the aggregation of the BIM BH3 peptide, which had been

stored at -80°C before the experiments. To address this possibility, we have repeated the experiments three times independently with freshly made BIM BH3, and the data is shown as new Fig. 1d. We have also replaced the comparison Fig. 1g with new data.

Fig 1F and Text. How are the Kd's calculated? Steady State or Kinetic analysis? To my eye kinetic analysis is not appropriate for these curves, certainly fitted curves have not been included so I presume the authors have not performed kinetic analysis. If Steady State then curves for steady state vs concentration should be provided to assess whether these are linear or not (i.e. appropriate for affinity analysis). Also, which curves were used in analysis. This is important given that some curves don't reach equilibrium (therefore can't be used for steady state), some curves show artefacts for the top concentrations, and for Bim there are different shaped curves for different concentrations indicating an artefact.

Response: We thank the Reviewer for this thoughtful comment. In the previous manuscript, the Kd's were calculated with Kinetic analysis. After carefully examining the curves, we agree with the Reviewer that Kinetic analysis is not appropriate for these curves. Accordingly, we have now analyzed these curves with Steady State analysis (and have explicitly indicated this in the Materials and Methods). These Steady State analyses are now shown in Fig. S1a-d. As indicated in Fig. S1a-d, most of the curves are included in the analysis, but the curves with artifacts are not included. Moreover, we have repeated the SPR assay for BAK/BIM BH3 interactions. In comparison to Kinetic analysis, the Steady State analyses yielded affinities about 2-fold lower for the BAK/BIM BH3, BAK/PUMA BH3 and BAK/HRK BH3 interactions, while the affinity of BAK/BMF BH3 was calculated to be 2-fold higher. As a result, we have reworded the conclusion to indicate that "BMF BH3 binds to BAK with higher affinity than BIM BH3, while HRK BH3 bind to BAK with similar affinity as PUMA BH3."

Given that many of the curves appear to be effected by artefact effects the authors should explain these experiments carefully. For example, what was used as a negative for these binding curves, importantly, how do these avoid effects from bulk shift.

Response: We used the blank channel as negative control. This was now indicated in the Materials and Methods. As the Reviewer suggested, we evaluated these curves now using steady state analysis, which will also help to avoid effects from bulk shift. A better negative control might be a mutant BAK protein. However, this BAK mutant would have to be different for BIM BH3 and BMF BH3. To avoid the confounding effect of having two different blank proteins, we have used blank channel as negative control.

Sup Fig 1 – the authors claim “Circular dichroism spectroscopy revealed similar degrees of α -helical-forming abilities of these BH3 peptides”. I am very surprised by the values presented in Sup Fig 1C based on the curves. BIK and HRK are clearly different to BMF in Supp fig 1b. Also, Bik and HRK do not look like classical CD spectra from samples predominately alpha helical.

Answer: The CD curves of these BH3 peptides in 30% TFE are not well fitted using the software to calculate the helicity, so we have now removed both the curves and the helicity calculated

for these peptides in 30% TFE (Fig. S2). This does not influence the conclusion that these BH3 peptides have similar helicity (we did not include TFE in the binding assays), and conclusion that the binding differences are not caused by differences in secondary structures.

Fig 2 C-F. Obatoclax has been shown to kill cells independent of Bax and Bak (Vogler et al., Cell Death and Diff, 2009), and its activity on pro-survival proteins is controversial. To inhibit the spectrum of pro-survivals a better combination would be navitoclax with one of the Mcl-1 specific inhibitors published by Abbvie or Servier.

Response: We have now used the MCL1 specific inhibitor S63845, as shown in Figs. 2d and S3b. Once again, both S63845 alone and S63845 in combination with navitoclax could not induce cytochrome c release from mitochondria of *Bax*^{-/-} MEFs.

Fig 3A and B – It is difficult to draw conclusions regarding Bik in these cells as the expression is very low.

Response: In the revised manuscript, we have observed better expression of BIK in Jurkat cells (Fig. 3a, b, c). Because we did not get good expression of BIK in MEFs, we have deleted BIK from the sentence describing conclusions from expression studies in MEFs.

Fig 3E – Bik should be included in this analysis.

Response: Because BIK is not expressed well in MEFs, we have performed this assay in Jurkat cells. In the revised manuscript, BIK is included as shown in Fig. 3d.

Fig 3G and F – I don't understand how the values for Annexin V positive cells are obtained. Eg. For Bik, it is listed as 51.1%, but the upper quadrants are 18.8% and 10.1% - 28.9%. Same is true for other FACS plots.

Response: We apologize to the Reviewer that we did not clearly describe how the quantitation was performed. In the revised manuscript we have tried to show this in Fig. 3a and 3b. What we are trying to indicate is the percentage of apoptotic cells (Annexin V+) in successfully transfected cells (EGFP+). This is calculated as the ratio of events in the *upper right* (cells positive for Annexin V and EGFP) to the sum of the *lower right* and *upper right* (cells positive for EGFP), which is also labeled at the right of dot blots. We hope that our change to the Legend of Figure 3 makes this clearer.

Fig 6B, D and G. Again, how were these affinities measured, steady state vs kinetic analysis, and based on the points raised above, is that analysis appropriate for these types of curves. In any case, the effect seems pretty clear cut. It could be represented differently to affinity, eg by max RU values, unless of course those RUs are effected by bulk shift or other artefact as discussed above.

Response: We thank for Reviewer for the suggestions. As the Reviewer pointed out, it is better

to use max RU values than Kd's in this situation. We have changed the figures to show maximum RU values, as shown in revised Figs. 6b, 6d and 6g in the manuscript.

All NMR plots – Graphs of chemical shift perturbations should be plotted as a function of BAK residue with the significance threshold used in fig 5 shown. This is standard for such analysis.

Response: We thank for the Reviewer for this helpful suggestion. We have now included three graphs of chemical shift perturbations as a function of BAK residues after PUMA BH3, BMF BH3 and HRK BH3 titration (Figure S6b, c and d). Indeed, after this analysis, we have found more residues on BAK that were significantly shifted after BMF BH3 and HRK BH3 titration, such as Q98, H99, F157, V158, M162, and L163, most of which are located in the $\alpha4$ - $\alpha6$ - $\alpha7$ groove. These updated results further support our conclusion that BMF BH3 and HRK BH3 interact on the $\alpha4$ - $\alpha6$ - $\alpha7$ groove of BAK. We have also now added these residues to the text (line 12-13, p. 11).

Reviewer #2 (Remarks to the Author):

Bax and Bak are two pore-forming proteins that can induce mitochondrial outer membrane permeabilization (MOMP) and apoptotic cell death. This activity is hence tightly regulated, mainly by the BH3 proteins that activate them and the antiapoptotic Bcl-2 family proteins that inhibit them. Unlike Bax that is a cytosolic protein in healthy cells, Bak is an integral membrane protein in the mitochondrial outer membrane (MOM). This implies that these two proteins might be activated via different mechanisms. One possibility is that different BH3 proteins are employed to activate Bax or Bak. Another possibility, not mutually exclusive from the first one, is that Bax and Bak are activated via different binding sites for BH3 proteins. Indeed, a binding site unique to Bax was found to be a binding site for BH3 proteins such as Bim and tBid to trigger Bax activation in the cytosol, which thereby was termed the trigger site. This trigger site, composed of the helices 1 and 6, is not the binding site for BH3 proteins Bim and tBid to activate Bak. Instead, a canonical BH3-binding site shared by Bak and Bax, also by all the antiapoptotic Bcl-2 proteins, is used for BH3 proteins Bim and tBid as well as Puma and Noxa to activate Bak in the MOM. Since this canonical BH3-binding site is blocked by the helix 9 in the cytosolic Bax protein, it would not be bound by a BH3 activator protein. However, once the soluble Bax protein inserts into the MOM using the helix 9, the canonical binding site would be free to accept a BH3 protein. According to another model, the binding of a BH3 protein to the canonical site is required for the Bax activation, like the Bak activation.

Now the Dai and Kaufmann groups provide evidence for a novel binding site in Bak for other BH3 proteins BMF and HRK to activate Bak. The evidence includes the SPR detected binding of the BMF and HRK BH3 peptides to Bak protein, the liposomal and mitochondrial membrane permeabilization induced by the BH3 peptides-activated Bak protein, and the apoptotic cell death induced by overexpression of BMF and HRK proteins that depended on Bak. Most importantly, the NMR based chemical shift perturbation data indicated a binding site composed of helices 4, 6 and 7 for the BMF and HRK BH3 peptides, which is different from the binding site indicated by

the control BH3 peptide from Puma that binds to the canonical site. Guided by the structural model, mutations in the helix 6 in the novel binding site reduced the binding of BMF and HRK, but not Bim, BH3 peptides to Bak protein. Moreover, mutations in the BMF and HRK BH3 peptides reduced the binding to Bak protein. These mutations, when introduced to the Bak or BMF and HRK proteins that were overexpressed in cells, inhibited apoptosis.

The conclusions made by the authors are well supported by solid experimental evidence. The discovery of the novel binding interaction for the two understudied BH3 proteins is significant not only to the basic understanding of how Bak is activated to induce cell death but to the therapeutic targeting of Bak to kill cancer cells.

Response: We thank the Reviewer for these positive comments.

The following issues need to be addressed before the acceptance of the manuscript for publication.

1. The activity of BMF and HRK proteins, as well as Bak, was only determined in the cells that overexpress them or their mutants. Can the authors repeat this assay in the cells whose apoptosis is dependent on the activation of endogenous BMF or HRK and Bak to determine whether the novel interaction is critical?

Response: We thank for the Reviewer for this suggestion. Previous study has indicated that anoikis (detachment-induced apoptosis) is dependent on BMF (Puthalakath H., et al., Science, 2001, 293: 1829-32). Therefore, we used this as a stimulus to assess the importance of the binding events we described. In new experiments that we have added to the manuscript, we first knocked down BMF using two different siRNAs in wild type MEFs and found that BMF knock-down could protect from anoikis (Fig. S7b). More importantly, anoikis is much lower in *Bak^{-/-}Bax^{-/-}* DKO MEFs reconstituted with BAK F161A (mutated on the $\alpha 6$ helix) compared cells reconstituted with wild type BAK (Fig. 7f), underscoring the importance of the interaction of BMF with the alternative site in a circumstance of endogenous BMF contributing to BAK activation. These results have been added to the revised manuscript (paragraph 3, p. 14).

2. The K_d values for HRK and BIK BH3 peptides binding to Bak protein are different by > 10-fold (Fig 1f), yet, the mitochondrial Cyto C release by these peptides is the same (Fig 2e). What is the explanation?

Response: Although BIK BH3 has a much weak affinity for BAK protein compared to HRK BH3 (Fig. 1f), BIK BH3 and HRK BH3 induce similar mitochondrial cyto C release from mitochondria of *Bax^{-/-}* MEFs. We think that cyto C release using mitochondria from *Bax^{-/-}* MEFs, which had classically been the way that the effect of Bak on cyto C release was studied, might only partially reflect the activation ability of these BH3 peptides. This is because Bak, after activation by these BH3 peptides, can still bind to BCL_{xL} or MCL1, which can prevent cytochrome C release. Because the BH3 peptides have different abilities to interact with BCL_{xL} or MCL1 to release activated BAK (Chen L, et al., Mol. Cell, 2005, 17, 393-403), this assay would

reflect both Bak activation ability of the BH3 peptides and their binding to BCL_{x_L} or MCL1 on these mitochondria. On the other hand, when we use another assay, where we put an excess of recombinant BAK and the BH3 peptides together in solution to cause cyto C release from mitochondria of *Bak*^{-/-}*Bax*^{-/-} MEFs (Fig. 2f), we found that HRK BH3 causes more cyto C release than BIK BH3, suggesting that HRK BH3 has a higher ability to activate BAK than BIK BH3. We think this assay might better reflect the activation ability of these BH3 peptides, because both the BH3 peptides and BAK used are in excess, which will minimize the differences in abilities of the BIK or HRK peptides to displace activated BAK from anti-apoptotic BCL2 proteins. We thank the reviewer for bringing this question up; we have added a few sentences in the discussion talking about this (paragraph 1, p. 16).

3. What are the concentrations of the BH3 peptides and Bak protein used in Fig 2f? In this figure, the Cyto C release activities of HRK and BIK BH3 peptides in the mitochondrial containing full-length Bak protein are different. In comparison, their activities in the mitochondria supplemented with a truncated Bak protein are the same. Why?

Response: The concentrations of the BH3 peptides and BAK protein used in Fig. 2f are 5 μM and 2 μM, respectively. This information is included in the Methods. The cyto C release activities of HRK and BIK BH3 peptides in the *Bax*^{-/-} mitochondria and *Bak*^{-/-}*Bax*^{-/-} mitochondria supplemented with recombinant BAK protein are different. While several explanations might be possible, an important factor is likely that both the BH3 peptides and the recombinant BAK protein are in excess in the latter experiment, which will minimize differences in abilities of these BH3 peptides to displace BAK from BCL_{x_L} and MCL1. Thus, we think the latter experiment better represents the BAK activation ability of the BH3 peptides than the former. This is discussed in the revised manuscript (paragraph 1, p. 16).

4. In line 173-174, “BIK...induced cell apoptosis in more than 75% of EGFP+ cells...” ; yet in Fig 3a, the percentage is ~40.

Response: We are sorry that this was not clearly described. What we tried to indicate in the figure (now Fig. 3a and 3b) was the percentage of successfully transfected cells (EGFP+) that are also Annexin V+, which is calculated by the ratio of events in the *upper right* to the sum of the *lower right* and *upper right*, also labeled at the right of dot blots. To make this clearer, we have changed the description in the legend to Figure 3.

5. Fig 3a, the right panel shows that BMF and BIK proteins were expressed at lower levels than other BH3 proteins. Thus, the apoptotic activities shown in the left panel need to be normalized by the protein levels in the cells. Similar issues with these or other proteins in Fig 3b to d, g, and h also need to be resolved.

Response: In the revised manuscript we no longer try to reach any conclusion about the ability of BIK to kill MEF cells because of the poor expression in MEF cells. Instead, we have worked to improve BIK expression in Jurkat cells and, in the revised manuscript, now show good expressions of BIK in these cells (now in Fig. 3a, b, c).

6. In Fig 5a, the perturbation of Bak structure by Puma BH3 peptide spreads into the novel site that was perturbed by the BMF or HRK peptide. The authors used the Bak structure bound by the Bid peptide to explain the cause without invoking the possibility that the Puma peptide may also bind the novel site. This is not convincing because the Puma peptide is different from the Bid peptide. Then in the functional assays shown in Fig 6 and 7, they used the Bim peptide and protein as the control, but not the Puma peptide and protein which they should use to validate the structural models. In particular, does the mutations in the novel site reduce the Puma to Bak, and the Puma induced membrane permeabilization and cell death by Bak?

Response: We thank the reviewer for pointing this out. After carefully examining the NMR chemical perturbation of BAK by PUMA BH3 (Table S1, and Figure 5a), we found several residues in the new site that have chemical shifts after the PUMA BH3 is added, including F157, F161, L163, H165, C166 and A168 (lines 3-5, p. 11). Therefore, we cannot rule out the possibility that PUMA BH3 has two binding sites on BAK. We have added a few sentences regarding this to the discussion (lines 3-8, p. 18).

For the functional assays, we have now added PUMA BH3 peptide in the BAK-mediated liposome release assay (Fig. 7a), the cell death assay (Fig. 7c) and also the PUMA induced BAK conformational change assay (Fig. 7d). These results indicated the mutation in the novel site does not reduce PUMA-induced BAK activation (Fig. 7d), membrane permeabilization (Fig. 7a) or cell death (Fig. 7c). These results have been included in the revised manuscript (paragraphs 3-4, p.13 and paragraph 1, p. 14).

7. All the mutations made for the novel site are in the helix 6. Since the site also composed of other helices, they should make other mutations, in particular, helix 4, to further confirm the structural model.

Response: Because helix $\alpha 4$ is also involved in the canonical binding, which involves the helices $\alpha 3$, $\alpha 4$ and $\alpha 5$, mutations in helix $\alpha 4$ would not discern the two different binding models.

8. The validate the cell assay data in Fig 7c, they need to show the expression levels of Bak wt and mutant protein.

Response: We have shown the expression levels of BAK wt and BAK F161A by western blot in Fig. 7b. We have tried to make this clearer in the figure legends.

Reviewed by Jialing Lin

Reviewer #3 (Remarks to the Author):

Ye et al. present a biochemical and structural investigation of BMF and HRK BH3-only proteins and pro-apoptotic BAK to demonstrate that BMF and HRK can directly activate BAX in vitro and in cells using an alternative binding site of BAK. This site is formed by $\alpha 4/\alpha 6/\alpha 7$ helices instead of

the canonical c-terminal groove that previous activators such as BID and BIM BH3 domains were found to bind. This is an interesting study in terms of the proposed alternative binding site of BAK, however BMF and HRK have been previously also shown that directly activate BAK. The authors need to provide more convincing case for the proposed binding site and direct activation of BAK.

The NMR data are not so clear. They need to be quantified across all residues and provide chemical shift perturbations plots for each peptide to appreciate significant chemical shifts changes in all residues of BAK and determine the binding site. It is not clear if all residues of the protein have been assigned which may be critical to the definition of the binding sites. Supplemental data in tables for assignments and residues undergoing chemical shift perturbations will be also helpful.

Response: To further test if BAK could be activated by BMF and HRK, we further generated *BIM^{-/-}PUMA^{-/-}BID^{-/-}NOXA^{-/-}BAK^{-/-}BAX^{-/-}* (HexaKO) Jurkat cells. Then we added a sgRNA-resistant inducible BAK construct in the HexaKO cells. This will at least exclude the possibility that BAK activation is induced by the known BAK activators such as BIM, PUMA, BID and NOXA. Moreover, BAK was induced to a much lower amount compared to endogenous levels (Fig. 3f) so that BAK would not be auto-activated and restrained by anti-apoptotic BCL2 paralogs as detected by immunoprecipitation assay (Fig. 3g). More importantly, under this condition, BMF and HRK still could induce cell apoptosis (Fig. 3h and 3i), further suggesting that BMF and HRK can directly activate BAK in the cells. This is far beyond the level of evidence provided for direct activation in early papers. These results have been added to the revised manuscript (last paragraph, p. 9 and first paragraph, p.10)

To further study BAK binding site by BMF and HRK BH3 peptides, we performed the simulation assays. As shown in Table S2, BMF and HRK BH3s are calculated to have much lower binding energies to the $\alpha4/\alpha6/\alpha7$ site than to the canonical BH3 binding groove, further supporting the idea that BMF and HRK BH3s bind to the $\alpha4/\alpha6/\alpha7$ site. (See last paragraph, p. 11 and first paragraph, p.12)

To make the NMR data clearer, we have now plotted chemical shift perturbations across all residues for each BH3 peptides (Figure S6b, c and d). We thank for the reviewer for this helpful suggestion. Indeed, after this analysis, we have found more residues on BAK that were significantly shifted after BMF BH3 and HRK BH3 titration, such as Q98, H99, F157, V158, M162, and L163, most of which sit in the $\alpha4-\alpha6-\alpha7$ groove, further supporting our conclusion that BMF BH3 and HRK BH3 interact with the $\alpha4-\alpha6-\alpha7$ groove of BAK. We have now added these residues to the text (lines 11-17, p. 11).

Finally, we have added a supplemental data table (Table S1), to include all the residues that exhibit significant chemical shift changes after PUMA BH3, BMF BH3 or HRK BH3 titrations. We have assigned 146 residues out of 169 residues of BAK (18-186, including 6 pro that do not have peaks on $^1\text{H}-^{15}\text{N}$ HSQC), which have also been added to the note of Table S2.

In addition, the alternative pocket is poorly defined by the presented data and the images in figure 5. Only a few residues are highlighted by NMR, how these residues comprise a binding site for these peptides? what are the residues of the groove and how the peptides can be predicted to interact with this proposed site? Is it feasible in terms of the interaction of each binding

partner?

Response: To make the residues involved in the noncanonical binding site clearer, we have performed a molecular dynamics simulation study of the interaction between BAK and BMF BH3 or HRK BH3. Through this study, we found BMF and HRK BH3 peptides showed much lower binding energies to the $\alpha 4$ - $\alpha 6$ - $\alpha 7$ groove compared to the canonical BH3 binding groove (Table S2), further supporting the idea that BMF and HRK BH3s interact with this new site. As indicated in the text (Figs 5d and 5e), binding of the BMF and HRK BH3 peptides induces conformational changes to form the hydrophobic groove at this $\alpha 4/\alpha 6/\alpha 7$ site. A close-up view showed that residue F161 on BAK interacts with the conserved Leu10 residues of the BH3 peptides (L37 in HRK and L137 in BMF), while H164 and H165 on BAK have close interactions with the conserved Asp15 from the two peptides (Figs. 5f and 5g). Moreover, the new $\alpha 4/\alpha 6/\alpha 7$ site is also hydrophobic (Fig. 5h), which is consistent with our mutation studies that suggest a mainly hydrophobic interaction between BAK and BMF and HRK BH3s.

The experiments in figures 2 and 3(except the liposomal assay, see below comment) do not discriminate between direct and indirect activation of apoptosis through BAK or BAX and this is not discussed in the text. Similar data have been also described in other publications regarding the apoptotic activity of overexpressed BH3-only proteins. I do not see any experimental condition that suggests that these proteins activate directly BAK in cells.

Response: The question of direct vs. indirect activation has received extensive prior attention. Some studies have provided evidence for direct activation and others have concluded that there is no direct activation. Indeed, at least two groups have published evidence on both sides of this argument without reconciling their conflicting claims. Thus, we are not certain that we can settle this question completely in the present manuscript.

However, to address this issue, we have engineered a new cell line derivative and performed additional experiments. As we mentioned in the answer for the first question, we have constructed and studied HexaKO Jurkat cells, which lack the well-established BH3 activators BIM, PUMA, BID, and NOXA (Fig. 3e). In addition, we also knocked out BAK and BAX, and inserted an inducible BAK expression vector. Under the conditions of our assay, there was no evidence for BAK autoactivation, i.e., no BAK pre-bound to BCL2, BCLX_L or MCL1 for a BH3-only protein to displace it (Fig. 3f and 3g). Under this condition, BMF and HRK still induced cell apoptosis (Fig. 3h and 3i), suggesting BMF and HRK directly activate BAK in cells. As described below, our experiments with BAK mutated in the $\alpha 4$ - $\alpha 6$ - $\alpha 7$ groove provide further support for the proposed direct activation.

The experiments in figure 7 show the potential of this novel region to regulate binding and activation of BAK-mediated pore activity and apoptosis. Authors should also test the mutants presented in the binding studies for the new site and the canonical BH3 domain and confirm how the canonical site mutants impair the activation. The should also provide biochemical data from the cell-based experiments that BAK is activated/oligomerized by BMF and HRK proteins from the specific interaction with the proposed site.

Response: We thank for the reviewer for this helpful suggestion. We have now tested BAK activation after expressing PUMA, BMF and HRK in cells. As shown in Figure 7d, PUMA induced apoptosis equally well in cells reconstituted with BAK F161A and WT BAK. In contrast, when BMF or HRK was expressed, apoptosis was markedly diminished in cells reconstituted with F161A BAK compared to WT BAK, consistent with our binding studies showing that the F161A substitution impairs binding of BMF BH3 or HRK BH3 to BAK. On the other hand, as the reviewer suggested, we also compared binding of anti-apoptotic BCL2 proteins to BAK F161A and WT BAK. If differences in killing were due to differences in displacement, the F161A mutation would be expected to result in a BAK protein that bound tighter to anti-apoptotic BCL2 paralogs. Instead, the F161A mutation did not affect interaction with anti-apoptotic proteins BCL2 paralogs (Fig. 7e), further supporting the idea that the differences in BAK activation are not due to displacement, but rather due to the direct activation abilities. These results have been added to the revised manuscript (paragraphs 1 and 2, p. 14)

Other comments:

The liposomal release mediated by BAK alone is high ~30%, do the authors use conditions and liposome composition that promote BAK auto-activation and BH3-mediated BAX activation?

Response: The liposomal release assays mediated by BAK alone, which has been used in our previous studies (Dai H., et al., *J. Cell Biol.* 2011, 194: 39-48; Dai H., et al., *Genes & Dev.* 2015, 29: 2140-2152) and also in papers from other groups (See references: Kuwana T., et al., *Mol. Cell*, 2005, 17: 525-535 ; Oh KJ. et al., *J. Biol. Chem.*, 2010, 285, 28924-28937; Du H., et al., *J. Biol. Chem.* 2011, 286, 491-501;), all yield a release of about 30% with BAK alone. Keeping in mind that the purpose of this *in vitro* assay is to assess whether BMF and HRK can promote BAK activation, a basal liposomal release at about 30% does not appear to be a problem as long as BMF and HRK increased the liposomal release significantly. On the other hand, because there is no anti-apoptotic BCL2 protein in this system, displacement BAK cannot happen under these conditions, yet BAK activation is enhanced by the BH3-only proteins. Thus, as suggested by the Reviewer, these results are consistent with direct activation.

In figure 3g, BIK is not expressed however there is significant apoptotic activity, how the authors can explain this discrepancy? Moreover, in figure 3C and 3D again there is very low expression of BIK proteins compared to other proteins but still the effect on Annexin V is significant and comparable to other BH3-only proteins.

Response: We thank for the reviewer pointing this out. As we now indicate in the text, BIK is not well expressed in the MEFs despite multiple attempts using several different transfection conditions. Because plasmids encoding the other BH3-only proteins are expressed well, we think the EGFP-BIK plasmid has been transfected into the MEF cells. However, due to some other mechanism (possibly a short half-life), much less BIK is seen on the western blot in MEFs (now shown in Figure S4). However, even this small amount of BIK could apparently induce MEF cell apoptosis. While BIK could also induce a lot of cell death in WT Jurkat cells, BIK induced much less cell death compared to BMF and HRK in the HexaKO Jurkat cells with

inducible BAK expression (Fig. 3h and 3i), suggesting a much weaker activation ability. We have added all these observations to the results (p.9).

It is surprising that Navitoclax and Obatoclax at these high doses are not able to induce cytochrome c release. Have the authors used a novel selective MCL-1 inhibitor such as S63845 or AM176 and its combination with Navitoclax?

Response: We thank for the reviewer for this suggestion and have now used the selective MCL-1 inhibitor S63845. In our mitochondria from *Bax*^{-/-} MEFs, neither S63845 alone (Fig. 2c) nor S63845 in combination with navitoclax (Fig. S3a) could release cytochrome c, suggesting BAK is not auto-activated in this cell line.

In summary, we have endeavored to address every comment by the three reviewers. We thank them for their helpful suggestions. We believe that the paper has been substantially improved through their efforts and hope that they will concur.

Reviewers' comments:

Reviewer #1 (Remarks to the Author):

My comments regarding an alternative explanation for the authors data, that binding could still be occurring at the canonical groove, has been mostly addressed. However, the phrasing “we could not totally exclude the possibility that BMF and HRK BH3s bind under certain conditions to the canonical BH3-binding groove for two reasons.” is a little disingenuous. It is in the experimental conditions tested within the paper that binding to the canonical groove is an alternative possibility, not “certain conditions”.

In regards to the peak drop then rise for sensograms for BAK with BMF, this occurs for both the 1600 nM and the 2500 nM binding curves, and as the authors note, is likely an artefact. They exclude the 2500 nM curve from their analysis, they should also exclude the 1600 nM analysis for Fig 6, or obtain data not affected by this artefact.

In regards to steady state analysis for binding affinities, there are unfortunately some issues with reporting affinity values. Determination of accurate affinities with steady state requires the top concentration to be at least 3 fold above the reported K_d . Unfortunately this is not reached for BIM and PUMA (Supp Figures 1c and d), so the authors can at best report an approximate range rather than reporting a precise value. For HRK (Sup 1b), the affinity is clearly outside the range of the experiment so at best the authors can report a value for the lower possible limit of the affinity value. The value for the affinity for the BMF is within the range of concentrations used within the experiment, but as noted above, I am concerned that the value for the 1600 nM injection is being influenced by artefact.

Are the interaction energies reported in Table S2 correct? My understanding is that typical binding energies for Bcl-2 family proteins with BH3s are in the range of -5 to -20 kcal/mol (eg see Jensen PNAS 2018 or Bhat J Mol Recog 2013).

I would note that the main hypotheses of the paper are not necessarily reliant on the reported values for the binding data discussed above, the authors should be able to find a way to report their data accurately and yet maintain the central postulate of the work.

Reviewer #2 (Remarks to the Author):

The authors have address all of my remarks satisfactory, except the following two

Reviewer 2's remark 2. The Kd values for HRK and BIK BH3 peptides binding to Bak protein are different by > 10-fold (Fig 1f), yet, the mitochondrial Cyto C release by these peptides is the same (Fig 2e). What is the explanation?

The authors' response: Although BIK BH3 has a much weak affinity for BAK protein compared to HRK BH3

(Fig. 1f), BIK BH3 and HRK BH3 induce similar mitochondrial cyto C release from mitochondria of Bax^{-/-} MEFs. We think that cyto C release using mitochondria from Bax^{-/-} MEFs, which had classically been the way that the effect of Bak on cyto C release was studied, might only partially reflect the activation ability of these BH3 peptides. This is because Bak, after activation by these BH3 peptides, can still bind to BCLxL or MCL1, which can prevent cytochrome C release. Because the BH3 peptides have different abilities to interact with BCLxL or MCL1 to release activated BAK (Chen L, et al., Mol. Cell, 2005, 17, 393-403), this assay would reflect both Bak activation ability of the BH3 peptides and their binding to BCLxL or MCL1 on these mitochondria.

Reviewer 2's remark 2a: If so, a prediction is that if one adds the small molecule inhibitors of Bcl-XL and Mcl-1 or the BH3 peptides that bind to Bcl-XI and Mcl-1 but not Bak, together with either BIK or HRK BH3 peptide, to the mitochondria from Bax KO MEFs, one will see that the BIK BH3 peptide is less active than the HRK BH3 peptide in release of cyto C. Can the authors do this experiment to validate their reasoning.

The authors' response: On the other hand, when we use another assay, where we put an excess of recombinant BAK and the BH3 peptides together in solution to cause cyto C release from mitochondria of Bak^{-/-}Bax^{-/-} MEFs (Fig. 2f), we found that HRK BH3 causes more cyto C release

than BIK BH3, suggesting that HRK BH3 has a higher ability to activate BAK than BIK BH3. We think this assay might better reflect the activation ability of these BH3 peptides, because both the BH3 peptides and BAK used are in excess, which will minimize the differences in abilities of the BIK or HRK peptides to displace activated BAK from anti-apoptotic BCL2 proteins. We thank the reviewer for bringing this question up; we have added a few sentences in the discussion talking about this (paragraph 1, p. 16).

Reviewer 2's remark 2b: In Line 361-362, the statement that "the HRK BH3 induced cytochrome c release in the presence of excess recombinant human BAK but not from Bax^{-/-} MEFs (Fig. 2e and 2f)" is not supported by the data in Fig. 2e and 2f.

5. Fig 3a, the right panel shows that BMF and BIK proteins were expressed at lower levels than other BH3 proteins. Thus, the apoptotic activities shown in the left panel need to be normalized by the protein levels in the cells. Similar issues with these or other proteins in Fig 3b to d, g, and h also need to be resolved.

The authors' response: In the revised manuscript we no longer try to reach any conclusion about the ability

of BIK to kill MEF cells because of the poor expression in MEF cells. Instead, we have worked to improve BIK expression in Jurkat cells and, in the revised manuscript, now show good expressions of BIK in these cells (now in Fig. 3a, b, c).

Reviewer 2's remark 5a: The expression level of BIK in Jurkat cells shown by the immunoblot in Fig. 3b is lower than other BH3 proteins, which is inconsistent with the flow cytometry data in Fig. 3a. Can the authors discuss this?

Jialing Lin

Reviewer #3 (Remarks to the Author):

In this revised manuscript the authors provided additional experiments such as biochemical experiments using BIM-/-PUMA-/-BID-/-NOXA-/-BAK-/-BAX-/- (HexaKO) Jurkat cells that improved the validation of the direct mechanism of BAK by BMF and HRK. However, the NMR analysis and interpretation of structural data is still no convincing at all that peptides can bind to an alternative site.

First of all, in this revision the authors present that PUMA BH3 helix induces additional chemical shifts from residues in a6 helix, which is part of the alternative proposed pocket. This information was missed from the previous analysis. Interestingly, the chemical shift plots per BAK residue from PUMA, BMF and HRK BH3 peptides that authors provide in this revision, show a very similar profile for the three peptides including that they also cause chemical shifts in residues of helix 1 which is not discussed at all in the text. The authors proposed that BMF and HRK BH3 peptides do not bind to the canonical groove because several residues that their peak lose intensity upon PUMA BH3 binding in the canonical pocket, seem not to be affected by the BMF and HRK peptides. However, close inspection of these residues suggest that there is reduction of the intensity, for example residues L97, L118, G126, F134 in BMF or HRK titrations. Not all residues that are mentioned in the text having disappearing cross peaks are shown in figure 4 and S6 and there is no calculation and comparison of cross peak intensities per BAK residue to demonstrate the authors claim. Perhaps the degree of cross peak reduction for the canonical site residues is not the same between PUMA and BMF or HRK but this could depend on the strength of the interaction and dissociation rate from BAK of each peptide. Nevertheless, the mapping of the chemical shift data as shown in figure 5 are not conclusive that there is an alternative pocket as in the case of PUMA and the canonical pocket. Interestingly, the authors suggest that PUMA may also bind to this alternative pocket based on the chemical shifts in helix a6 although in the functional assays the use PUMA BH3 binding and activation to support the canonical pocket-mediated activation as PUMA-mediated BAK activation and apoptosis is not affected by F161A mutant. This is even more confusing how this could happen. From these data, a very likely explanation is that BMF and HRK BH3 peptides bind to the canonical pocket with somewhat different orientation or bend in their structure from PUMA and BIM BH3 peptides that can push the helix a4 towards helix a6 in a different manner. Such conformational change of helix a4 towards a6 has been also seen in NMR and crystal structures of BID and BIM BH3 helices (Moldoveanu T et al, 2013, Brouwer JM et al 2017). This would also be consistent with the differences seen with BAK mutants evaluated. Moreover, the authors present figures of BAK structure from molecular dynamics simulation, suggesting that peptides can bind to an alternative pocket however the analysis is superficial focusing on two residues and not showing data from the simulations for both pockets, distances of the interactions over time of the simulations, conformations of the peptide etc. The figures 5f and 5g for example show that F161 is not in close distance with the L137 or L37 and one wonders how the mutation of F161 with a hydrophobic residue Ala would make such significant loss of binding and activation from the BMF and HRK BH3s. An alternative explanation is that F161 plays a role in the conformational change induced by BMF and HRK through the helix a4 (eg Y108) – helix a6 (eg H165) interaction.

Reviewer #1 (Remarks to the Author):

My comments regarding an alternative explanation for the authors data, that binding could still be occurring at the canonical groove, has been mostly addressed. However, the phrasing “we could not totally exclude the possibility that BMF and HRK BH3s bind under certain conditions to the canonical BH3-binding groove for two reasons.” is a little disingenuous. It is in the experimental conditions tested within the paper that binding to the canonical groove is an alternative possibility, not “certain conditions”.

Response: We regret the misstatement and have deleted that sentence. Following the suggestions from the Editor, we have analyzed our NMR data in greater detail, performed more comprehensive simulations and obtained new results suggesting that BMF and HRK BH3s bind to both grooves, whereas BIM BH3 binds only to the canonical groove (Table S2). Now we have added a new paragraph on pages 21-22 discussing the binding of BMF and HRK BH3s to both grooves of BAK.

In regards to the peak drop then rise for sensograms for BAK with BMF, this occurs for both the 1600 nM and the 2500 nM binding curves, and as the authors note, is likely an artefact. They exclude the 2500 nM curve from their analysis, they should also exclude the 1600 nM analysis for Fig 6, or obtain data not affected by this artefact.

Response: We have conducted the SPR analysis of BAK interacting with the BMF domain BH3 three additional times. By constituting the BMF BH3 at a higher stock concentration, we could eliminate the problem with the peak drop in these experiments (See Figures 1a, 6c and 6f). The indicated panels been replaced with the new results.

In regards to steady state analysis for binding affinities, there are unfortunately some issues with reporting affinity values. Determination of accurate affinities with steady state requires the top concentration to be a at least 3 fold above the reported K_d. Unfortunately this is not reached for BIM and PUMA (Supp Figures 1c and d), so the authors can at best report an approximate range rather than reporting a precise value. For HRK (Sup 1b), the affinity is clearly outside the range of the experiment so at best the authors can report a value for the lower possible limit of the affinity value. The value for the affinity for the BMF is within the range of concentrations used within the experiment, but as noted above, I am concerned that the value for the 1600 nM injection is being influenced by artefact.

Response: We have changed Figure 1f and its figure legend to report an approximate range of K_ds. We have also changed the description in the text (see last paragraph of page 5). In addition, as we indicated in the last response, we have conducted the SPR experiment using BAK with BMF BH3 again.

Are the interaction energies reported in Table S2 correct? My understanding is that typical binding energies for Bcl-2 family proteins with BH3s are in the range of -5 to -20 kcal/mol (eg see Jensen PNAS 2018 or Bhat J Mol Recog 2013).

Response: The interaction energies in Table S2 were correct. We apologize for the misleading wording in the text of Line 265. It should have read “with much lower interaction energies” rather than “with much lower binding energies.” As stated in Table S2, these were calculated interaction energies (i.e., relative enthalpy energies) and offer insight into which binding groove is energetically favored. However, they are not comparable to the reported binding energies, which are absolute free energies of binding.

In the current version, we have replaced the relative interaction energies in Table S2 with the occurrence of the most populated conformer. The analysis reflects the stability of the conformer and is a better proxy of the relative free energy of the conformer.

I would note that the main hypotheses of the paper are not necessarily reliant on the reported values for the binding data discussed above, the authors should be able to find a way to report their data accurately and yet maintain the central postulate of the work.

Response: We greatly appreciate the reviewer’s helpful suggestions and have made the changes described above.

Reviewer #2 (Remarks to the Author):

The authors have address all of my remarks satisfactory, except the following two

Reviewer 2’s remark 2. The Kd values for HRK and BIK BH3 peptides binding to Bak protein are different by > 10-fold (Fig 1f), yet, the mitochondrial Cyto C release by these peptides is the same (Fig 2e). What is the explanation?

The authors’ response: Although BIK BH3 has a much weak affinity for BAK protein compared to HRK BH3 (Fig. 1f), BIK BH3 and HRK BH3 induce similar mitochondrial cyto C release from mitochondria of Bax-/- MEFs. We think that cyto C release using mitochondria from Bax-/- MEFs, which had classically been the way that the effect of Bak on cyto C release was studied, might only partially reflect the activation ability of these BH3 peptides. This is because Bak, after activation by these BH3 peptides, can still bind to BCLxL or MCL1, which can prevent cytochrome C release. Because the BH3 peptides have different abilities to interact with BCLxL or MCL1 to release activated BAK (Chen L, et al., Mol. Cell, 2005, 17, 393-403), this assay would reflect both Bak activation ability of the BH3 peptides and their binding to BCLxL or MCL1 on these mitochondria.

Reviewer 2’s remark 2a: If so, a prediction is that if one adds the small molecule inhibitors of Bcl-XL and Mcl-1 or the BH3 peptides that bind to Bcl-Xl and Mcl-1 but not Bak, together with either BIK or HRK BH3 peptide, to the mitochondria from Bax KO MEFs, one will see that the BIK BH3 peptide is less active than the HRK BH3 peptide in release of cyto C. Can the authors do this experiment to validate their reasoning.

Response: We thank the Reviewer for this helpful suggestion. In response, we have performed additional assays (Figs. 2f and S3c). When the MCL1 inhibitor S63845 was added together with BIK or HRK BH3 peptide to the mitochondria from *Bax*^{-/-} MEFs, the BIK BH3 peptide was about 10-fold less active than the HRK BH3 peptide (Fig. 2f), in agreement with the binding affinities shown in Fig. 1 and the liposome release assay shown in Fig. 2a-b. In contrast, when the BCL2/BCL_{xL} inhibitor navitoclax was added together with BIK or HRK BH3 peptide, it did not increase the efficiency of cytochrome c release induced by either BIK BH3 or HRK BH3 (Fig. S3c), suggesting that BAK after activation is predominantly bound to MCL1 on mitochondria from *Bax*^{-/-} MEFs, but not BCL2 or BCL_{xL}. All these results have been indicated in the Results (last full paragraph on p. 7) and Discussion (second full paragraph, p. 18) of the revised manuscript.

The authors' response: On the other hand, when we use another assay, where we put an excess of recombinant BAK and the BH3 peptides together in solution to cause cyto C release from mitochondria of Bak^{-/-}Bax^{-/-} MEFs (Fig. 2f), we found that HRK BH3 causes more cyto C release than BIK BH3, suggesting that HRK BH3 has a higher ability to activate BAK than BIK BH3. We think this assay might better reflect the activation ability of these BH3 peptides, because both the BH3 peptides and BAK used are in excess, which will minimize the differences in abilities of the BIK or HRK peptides to displace activated BAK from anti-apoptotic BCL2 proteins. We thank the reviewer for bringing this question up; we have added a few sentences in the discussion talking about this (paragraph 1, p. 16).

Reviewer 2's remark 2b: In Line 361-362, the statement that "the HRK BH3 induced cytochrome c release in the presence of excess recombinant human BAK but not from Bax^{-/-} MEFs (Fig. 2e and 2f)" is not supported by the data in Fig. 2e and 2f.

Response: We thank for the Reviewer for pointing this out. The statement "the HRK BH3 induced cytochrome c release in the presence of excess recombinant human BAK but not from *Bax*^{-/-} MEFs" has been changed on p. 18 of the revised manuscript to "the HRK BH3 has a higher ability than BIK BH3 to induce cytochrome c release in the presence of excess recombinant human BAK, while the ability to induce cytochrome c release from *Bax*^{-/-} MEFs is no different for the two peptides." As indicated above, we then go on to address the discrepancy between the assays by performing the release assay in S63845 or navitoclax as suggested by the Reviewer and conclude that the discrepancy appears to reflect binding by anti-apoptotic proteins, specifically MCL1.

5. Fig 3a, the right panel shows that BMF and BIK proteins were expressed at lower levels than other BH3 proteins. Thus, the apoptotic activities shown in the left panel need to be normalized by the protein levels in the cells. Similar issues with these or other proteins in Fig 3b to d, g, and h also need to be resolved.

The authors' response: In the revised manuscript we no longer try to reach any conclusion about the ability of BIK to kill MEF cells because of the poor expression in MEF cells. Instead, we have worked to improve BIK expression in Jurkat cells and, in the revised manuscript, now show good expressions of BIK in these cells (now in Fig. 3a, b, c).

Reviewer 2's remark 5a: The expression level of BIK in Jurkat cells shown by the immunoblot in Fig. 3b is lower than other BH3 proteins, which is inconsistent with the flow cytometry data in Fig. 3a. Can the authors discuss this?

Response: We think the low expression level of BIK in Jurkat cells might reflect a short half-life. To test this hypothesis, we performed cycloheximide chase experiments. 24 hours after EGFP-tagged BIK, NOXA (a protein known to have a short half-life) or empty vector was transfected into Jurkat cells, the half-lives of the EGFP-tagged protein were assessed by inhibiting protein translation using 30 $\mu\text{g/ml}$ cycloheximide and performing flow microfluorimetry at periodic intervals. As shown in the figure below, EGFP-BIK exhibited a half-life of about 3.5 hours, less than that of EGFP-NOXA (~ 6 h) in Jurkat cells, providing an explanation for the low expression level of BIK in Jurkat cells compared to other BH3 proteins. We have added this possible explanation to the Discussion (first paragraph on p. 19).

Figure A. EGFP-BIK has a short half-life. Jurkat cells were transfected with EGFP, EGFP-BIK, or EGFP-NOXA, incubated in the presence of 10 μM Q-VD-Oph for 24 hours, and then treated with 30 $\mu\text{g/ml}$ cycloheximide for the indicated length of time. Starting from the time of cycloheximide addition, the cells were harvested at the indicated time point, assayed for EFGP fluorescence by flow cytometry, and the percentage of the mean fluorescence of the indicated time point to the initial value were estimated. Error bars: mean \pm S.D. of three independent experiments.

In summary, we appreciate the questions raised by Reviewer #2 and have endeavored to address them by performing additional experiments and making changes to the Results and Discussion.

Reviewer #3 (Remarks to the Author):

In this revised manuscript the authors provided additional experiments such as biochemical experiments using BIM-/-PUMA-/-BID-/-NOXA-/-BAK-/-BAX-/- (HexaKO) Jurkat cells that improved the validation of the direct mechanism of BAK by BMF and HRK. However, the NMR analysis and interpretation of structural data is still no convincing at all that peptides can bind to an alternative site.

Response: We thank for the reviewer for acknowledging our efforts to show direct activation of BAK by BMF and HRK.

First of all, in this revision the authors present that PUMA BH3 helix induces additional chemical shifts from residues in $\alpha 6$ helix, which is part of the alternative proposed pocket. This information was missed from the previous analysis. Interestingly, the chemical shift plots per BAK residue from PUMA, BMF and HRK BH3 peptides that authors provide in this revision, show a very similar profile for the three peptides including that they also cause chemical shifts in residues of helix 1 which is not discussed at all in the text.

Response: We have now added a paragraph discussing the chemical shifts induced in $\alpha 1$ of BAK by PUMA, BMF and HRK BH3 peptides. As indicated in the first paragraph of p. 21, the chemical shift changes in helix $\alpha 1$ are thought to reflect the conformational change that occurs at $\alpha 1$ whenever BAK is activated.

The authors proposed that BMF and HRK BH3 peptides do not bind to the canonical groove because several residues that their peak lose intensity upon PUMA BH3 binding in the canonical pocket, seem not to be affected by the BMF and HRK peptides. However, close inspection of these residues suggest that there is reduction of the intensity, for example residues L97, L118, G126, F134 in BMF or HRK titrations. Not all residues that are mentioned in the text having disappearing cross peaks are shown in figure 4 and S6 and there is no calculation and comparison of cross peak intensities per BAK residue to demonstrate the authors claim. Perhaps the degree of cross peak reduction for the canonical site residues is not the same between PUMA and BMF or HRK but this could depend on the strength of the interaction and dissociation rate from BAK of each peptide. Nevertheless, the mapping of the chemical shift data as shown in figure 5 are not conclusive that there is an alternative pocket as in the case of PUMA and the canonical pocket.

Response: We thank for the Reviewer for providing the opportunity to expand on this point. We have now added the analysis of the reduction of intensity for the NMR perturbation data of all three peptides (as shown in supplemental Fig. S6b-d, right). In the previous version of the manuscript, we used the criterion of “complete” peak loss in the NMR perturbation spectrum to define residues involved in peptide binding at a ratio of peptide: BAK=4:1 (Peak loss means the peak intensity decreased to the level of noise, therefore it is hard to define the peak. It was displayed as light blue in Fig. S6b.). Although some previous studies also used combined chemical shift (including chemical shifts and peak disappearance) to identify residues involved in ligand interaction sites by NMR,

usually only the peak disappearance is considered (For example: *Dominguez C, et al., Nucleic Acids Res., 2006, 34: 3634-45, <https://academic.oup.com/nar/article/34/13/3634/1157734>; Tossavainen H, et al., J. Biol. Chem., 2016, 291: 16307-17, <http://www.jbc.org/content/291/31/16307.short>; Huang X, et al., J. Phys. Chem. B, 2012, 116: 14235-44, <https://pubs.acs.org/doi/10.1021/jp308207h>).*

As indicated in the figures now (and also as the Reviewer mentioned), some of the residues in the canonical binding groove, including L97, L118, G126, and F134, also showed some degree of reduction in the peak intensity after BMF or HRK BH3 perturbation. As indicated in the revised manuscript, these observations raise the hypothesis that BMF BH3 and HRK BH3 can bind to both grooves on BAK: The canonical BH3 binding groove and the alternative $\alpha 4/\alpha 6/\alpha 7$ groove. To further evaluate this possibility, we performed extensive MD simulations and obtained new data suggesting that BMF and HRK BH3s bind to both grooves, whereas BIM BH3 binds to the canonical BH3 binding groove only (Table S2). To reflect these findings, we have substantially revised our manuscript.

Interestingly, the authors suggest that PUMA may also bind to this alternative pocket based on the chemical shifts in helix $\alpha 6$ although in the functional assays the use PUMA BH3 binding and activation to support the canonical pocket-mediated activation as PUMA-mediated BAK activation and apoptosis is not affected by F161A mutant.

Response: PUMA-induced NMR perturbation in the residues of $\alpha 6$ might reflect two processes: 1) PUMA might also bind to the alternative pocket and 2) the $\alpha 4$ helix of BAK after PUMA BH3 binding might move towards $\alpha 6$ and induce the chemical shift changes of these residues (Brouwer JM, et al. *Mol. Cell*, 2017, 68:659-672; Moldoveanu T, et al. *Nat. Struct. Mol. Biol.*, 2013, 20: 589-597). Our further study indicates that the F161A mutation did not affect activation by PUMA BH3 (Fig. 7a). However, this cannot rule out the possibility that PUMA BH3 binds to two grooves, with a predominant role for PUMA BH3 binding to the canonical BH3 binding groove. Accordingly, in the Discussion (p. 20) we have changed the explanation to "...suggesting the possibility that PUMA BH3 also binds both grooves, but mostly to the canonical groove because the F161A mutation did not affect BAK activation by PUMA BH3. We could not rule out the possibility that addition of PUMA BH3 causes BAK $\alpha 4$ to move towards $\alpha 6$ as an explanation for some of the observed chemical shift perturbations."

This is even more confusing how this could happen. From these data, a very likely explanation is that BMF and HRK BH3 peptides bind to the canonical pocket with somewhat different orientation or bend in their structure from PUMA and BIM BH3 peptides that can push the helix $\alpha 4$ towards helix $\alpha 6$ in a different manner. Such conformational change of helix $\alpha 4$ towards $\alpha 6$ has been also seen in NMR and crystal structures of BID and BIM BH3 helices (Moldoveanu T et al, 2013, Brouwer JM et al 2017). This would also be consistent with the differences seen with BAK mutants evaluated.

Response: There might be two explanations for the PUMA-induced NMR chemical shift changes: i) As the reviewer suggested, $\alpha 4$ helix of BAK might move towards $\alpha 6$ after PUMA

BH3 binding, and ii) PUMA BH3 might bind to both the canonical BH3 binding groove and the $\alpha 4/\alpha 6/\alpha 7$ groove. However, even if the PUMA-induced NMR chemical shift change is due to the first explanation, it is still different from BMF and HRK BH3s based on three observations.

First, the reviewer advances the hypothesis that the degree of cross peak reduction for the canonical site residues might be different for PUMA vs. BMF or HRK because of differences in the strength of the interaction and dissociation rate from BAK of each peptide. As we measured in Fig. 1, the equilibrium dissociation constant for BAK to BMF BH3 is even smaller (tighter) than PUMA BH3, while the HRK BH3 shows similar equilibrium dissociation constant to BAK as PUMA BH3. Therefore, the fact that the degree of cross peak reduction for the canonical site residues is lower in BMF or HRK compared to PUMA, is not because of worse affinity. Instead, we favor the alternative explanation that more BMF and HRK BH3 peptide binds to the alternative site, which was also in agreement with the NMR perturbation in the $\alpha 4/\alpha 6/\alpha 7$ site, i.e., that more residues (such as F161) showed chemical shifts in the $\alpha 6/\alpha 7$ area for BMF and HRK BH3s than for PUMA BH3.

Second, our mutation data further supported the idea that BMF and HRK BH3 peptides interact with the $\alpha 4/\alpha 6/\alpha 7$ groove to induce BAK activation. As the mutation and functional assay indicated, the F161A mutation and H164/H165 mutation of BAK only affect the binding of BMF and HRK BH3s, but not the binding of BIM BH3. Furthermore, the F161A mutation affects both BMF- and HRK-induced liposome release and cell death, but not liposome release and cell death induced by BIM and PUMA. Therefore, these data suggest BMF and HRK bind to the $\alpha 4/\alpha 6/\alpha 7$ groove.

Finally, following the suggestion of the Editor and reviewer, we have performed more comprehensive simulations and obtained new data suggesting that BMF and HRK BH3s can potentially bind to both grooves, whereas BIM BH3 binds to the canonical groove only (Table S2).

Moreover, the authors present figures of BAK structure from molecular dynamics simulation, suggesting that peptides can bind to an alternative pocket however the analysis is superficial focusing on two residues and not showing data from the simulations for both pockets, distances of the interactions over time of the simulations, conformations of the peptide etc.

Response: In the new Table S2, we have provided detailed results from a rigorous analysis of comprehensive, microsecond, isobaric–isothermal MD simulations of BAK and the F161A mutant in complex with BMF, HRK, and BIM at all grooves except the simulation of BIM at the noncanonical groove of F161A (because BIM does not bind to the noncanonical groove of the wildtype). Specifically, for each of the three peptides (BMF, HRK, and BIM) binding at both grooves, we have provided 1) the distances and conformations that are averaged from 20,000 conformations from 20 independent MD simulations, each of which lasted 316 ns, and 2) the occurrences of the top three-most populated conformation clusters to demonstrate the strength of the peptide binding, as strong binding is typically signified by a large occurrence of the first-most populated

cluster followed by much smaller occurrences of the second- and third-most populated clusters. Further, we have provided the results of our control studies at two different temperatures in which miniprotein CLN025 initially adopted a fully extended linear conformation and subsequently folded autonomously into a beta hairpin structure, as captured in the first-most populated conformation cluster, with an alpha carbon root mean square deviation of 1.7 Å from the corresponding experimentally determined structure. The simulation data in Table S2 are in excellent agreement with our NMR data.

The figures 5f and 5g for example show that F161 is not in close distance with the L137 or L37 and one wonders how the mutation of F161 with a hydrophobic residue Ala would make such significant loss of binding and activation from the BMF and HRK BH3s. An alternative explanation is that F161 plays a role in the conformational change induced by BMF and HRK through the helix $\alpha 4$ (eg Y108) – helix $\alpha 6$ (eg H165) interaction.

Response: As shown by new simulations of BMF and HRK BH3 peptides binding to BAK F161A (Table S2), replacement of the relatively bulky phenyl group with a methyl group through the F161A mutation shrinks both the hydrophobic cavity (at which L137 or L37 docks) and the $\alpha 4/\alpha 6/\alpha 7$ groove region (where the pi helical segment of BMF and HRK binds). Because $\alpha 4$ separates the $\alpha 4/\alpha 6/\alpha 7$ and canonical grooves, the $\alpha 4/\alpha 6/\alpha 7$ groove contraction results in expansion of the canonical groove and thus weakens the binding of the two peptides at the canonical groove as well. Therefore, the F161A mutation causes significant losses of binding and activation from the two peptides. We have now added this explanation to the current revision.

In summary, we thank the three Reviewers for their insightful comments and suggestions, which we have addressed through a combination of additional wet bench experiments, additional simulations and more judicious editing. We feel that these changes have strengthened the manuscript and hope that the Reviewers and Editor will concur.

Reviewers' comments:

Reviewer #1 (Remarks to the Author):

Unfortunately it is still my opinion that the experimental data can be explained through binding to the canonical groove, the newly presented data do not exclude this possibility, and if anything add further confusion to the manuscript. For example, the new modeling has HRK preferring the canonical – “3) HRK binds to both grooves with a preference for the canonical groove”, this points to inconsistency between their experimental data and the modeling, the modeling suggest a preference of HRK to the canonical groove, but the chemical shifts, and other data, have HRK more like BMF than BIM or PUMA. I also don't understand why the modeling with the F161A mutant excludes the possibility, in fact that modeling suggests that the mutation changes the nature of the canonical groove, as I had previously suggested could happen, therefore effects could be through that canonical groove and through differences in the binding mode for the different peptides binding to that groove as proposed by reviewer 3. There is no definitive evidence shown that the effects are not happening through the canonical groove, as such I don't believe this work can be published without a genuine discussion of the alternative possibility as presented in my previous reviews.

Reviewer #2 (Remarks to the Author):

The authors have satisfactorily addressed my requests by performing additional experiments. This re-revised manuscript, in my opinion, can be accepted for publication.

Jialing Lin

Reviewer #3 (Remarks to the Author):

The authors have revised their manuscript and provide additional data for the proposed differences in binding and activating BAK by BH3-only proteins previously established e.g. Bim, Bid, Noxa, and additional such as Bmf, Hrk and Bik. The new manuscript provides clarification on the binding experiments, and biochemical data using liposomal and mitochondrial release assays. Moreover the NMR analysis of different BH3-only peptides has been improved compared to the initial submission, and extensive molecular dynamic data are supportive of the proposed mechanism.

The results and discussion regarding the finding that an additional surface outside the canonical BH3-only pocket is also involved with the binding and activation of BAK by BMF and HRK proteins, are more balanced and indicate alternative conclusions. Functional and mutagenesis data are straightforward and indicate the differences among the BH3-only proteins. Some NMR data are open to alternative interpretation and authors should make sure the text provide an alternative interpretation. One thing that I find challenging is how that MD data are presented in table 2. I would encourage the authors to show in the supplement, poses from the simulations before, during and at the end of simulation and potentially highlight the conformational changes that are discussed.

In response to the comments of the Reviewers, additional data have been added as requested, the entire manuscript has been edited, and the discussion extensively revised. Specific comments have been addressed as follows:

Reviewer #1 (Remarks to the Author):

Unfortunately it is still my opinion that the experimental data can be explained through binding to the canonical groove, the newly presented data do not exclude this possibility, and if anything add further confusion to the manuscript. For example, the new modeling has HRK preferencing the canonical – “3) HRK binds to both grooves with a preference for the canonical groove”, this points to inconsistency between their experimental data and the modeling, the modeling suggest a preference of HRK to the canonical groove, but the chemical shifts, and other data, have HRK more like BMF than BIM or PUMA. I also don't understand why the modeling with the F161A mutant excludes the possibility, in fact that modeling suggests that the mutation changes the nature of the canonical groove, as I had previously suggested could happen, therefore effects could be through that canonical groove and through differences in the binding mode for the different peptides binding to that groove as proposed by reviewer 3. There is no definitive evidence shown that the effects are not happening through the canonical groove, as such I don't believe this work can be published without a genuine discussion of the alternative possibility as presented in my previous reviews.

Response: In light of these new comments, we have re-written the Discussion and revised the other sections to clarify that: 1) We have experimental and computational data collectively indicating that BMF and HRK can bind to either the canonical or noncanonical groove; 2) we do not have data to exclude the binding of BMF or HRK at the canonical groove that allosterically perturbs the residues at the noncanonical groove, a point that Reviewer 1 has been suggesting all along and that is consistent with our simulations of the F161A mutant as noted above by Reviewer 1 as well; and 3) binding of BMF or HRK at the noncanonical groove for BAK activation remains one of several models that could explain the data .

We have also clarified in the present revision that both HRK and BIM prefer the canonical groove (74% for HRK and 75% for BIM) over the noncanonical groove (53% for HRK and 35% for BIM), but HRK is more like BMF than BIM because HRK still binds to some extent the noncanonical site (53%) and BIM binds the noncanonical site less (35%). Therefore, there is no inconsistency between our experimental and computational data.

In the present revision we have also added new data (Fig. 7d and S8b) showing that BMF and HRK can still induce BAK-mediated cell death, while BIM cannot, when the canonical groove is blocked by mutations (BAK G126S/N86G, where the G126S mutation irreversibly blocks the groove and the BAK N86G mutation is a reciprocal mutation used to allow BAK oligomerization, see G. Dewson *et al.*, *Mol. Cell*, 2008, 30, 369-380). This addition makes our alternative groove hypothesis more plausible, but we would like to emphasize that the

addition still does not minimize the allosteric perturbation possibility because of our simulation results indicating that BMF and HRK can bind to either of the two grooves.

Reviewer #2 (Remarks to the Author):

The authors have satisfactorily addressed my requests by performing additional experiments. This re-revised manuscript, in my opinion, can be accepted for publication.

Response: We thank the Reviewer for the positive opinion.

Reviewer #3 (Remarks to the Author):

The authors have revised their manuscript and provide additional data for the proposed differences in binding and activating BAK by BH3-only proteins previously established e.g. Bim, Bid, Noxa, and additional such as Bmf, Hrk and Bik. The new manuscript provides clarification on the binding experiments, and biochemical data using liposomal and mitochondrial release assays. Moreover the NMR analysis of different BH3-only peptides has been improved compared to the initial submission, and extensive molecular dynamic data are supportive of the proposed mechanism.

Response: We thank for the reviewer for the positive comments.

The results and discussion regarding the finding that an additional surface outside the canonical BH3-only pocket is also involved with the binding and activation of BAK by BMF and HRK proteins, are more balanced and indicate alternative conclusions. Functional and mutagenesis data are straightforward and indicate the differences among the BH3-only proteins. Some NMR data are open to alternative interpretation and authors should make sure the text provide an alternative interpretation.

Response: We have extensively revised the Discussion in an effort to present a more balanced view that acknowledges that alternative explanations for the data are possible and further experiments warranted. See our response to Reviewer 1.

One thing that I find challenging is how that MD data are presented in table 2. I would encourage the authors to show in the supplement, poses from the simulations before, during and at the end of simulation and potentially highlight the conformational changes that are discussed.

Response: We thank for the Reviewer for pointing this out. We have now added Supplementary Figure 7 to expand on the data in Table S2 by showing conformations of all peptides listed in Table S2 to highlight conformational differences among different peptides at different grooves and conformational differences before and at the end of the simulations.

In summary, we thank the three Reviewers for their insightful comments and suggestions, which we have addressed through a combination of additional wet bench experiments, and

extensive editing. We feel that these changes have strengthened the manuscript and hope that the Reviewers and Editor will concur.